

# Exact results for duality-covariant integrated correlators in $\mathcal{N} = 4$ SYM with general classical gauge groups

**Daniele Dorigoni[1], Michael B. Green[2,3] and Congkao Wen[3]**

**1** Centre for Particle Theory & Department of Mathematical Sciences, Durham University, Lower Mountjoy, Stockton Road, Durham DH1 3LE, UK
**2** Department of Applied Mathematics and Theoretical Physics, Wilberforce Road, Cambridge CB3 0WA, UK
**3** Centre for Theoretical Physics, Department of Physics and Astronomy, Queen Mary University of London, London, E1 4NS, UK

## Abstract

We present exact expressions for certain integrated correlators of four superconformal primary operators in the stress tensor multiplet of $\mathcal{N} = 4$ supersymmetric Yang–Mills (SYM) theory with classical gauge group, $G_N = SO(2N)$, $SO(2N + 1)$, $USp(2N)$. These integrated correlators are expressed as two-dimensional lattice sums by considering derivatives of the localised partition functions, generalising the expression obtained for $SU(N)$ gauge group in our previous works. These expressions are manifestly covariant under Goddard-Nuyts-Olive duality. The integrated correlators can also be formally written as infinite sums of non-holomorphic Eisenstein series with integer indices and rational coefficients. Furthermore, the action of the hyperbolic Laplace operator with respect to the complex coupling $\tau = \theta/(2\pi) + 4\pi i/g_{YM}^2$ on any integrated correlator for gauge group $G_N$ relates it to a linear combination of correlators with gauge groups $G_{N+1}$, $G_N$ and $G_{N-1}$. These "Laplace-difference equations" determine the expressions of integrated correlators for all classical gauge groups for any value of $N$ in terms of the correlator for the gauge group $SU(2)$. The perturbation expansions of these integrated correlators for any finite value of $N$ agree with properties obtained from perturbative Yang–Mills quantum field theory, together with various multi-instanton calculations which are also shown to agree with those determined by supersymmetric localisation. The coefficients of terms in the large-$N$ expansion are sums of non-holomorphic Eisenstein series with half-integer indices, which extend recent results and make contact with low order terms in the low energy expansion of type IIB superstring theory in an $AdS_5 \times S^5/\mathbb{Z}_2$ background.


# 1 Introduction and outline

In [1,2] an integrated correlator of four superconformal primary operators in the stress tensor multiplet of $\mathcal{N} = 4$ supersymmetric $SU(N)$ Yang–Mills (SYM) theory was expressed as a two-dimensional lattice sum that is manifestly invariant under $SL(2,\mathbb{Z})$ Montonen-Olive duality and is valid for all values of $N$ and the coupling constant $\tau = \tau_1 + i\tau_2 = \theta/(2\pi) + i4\pi/g_{YM}^2$ in the upper-half plane $\tau_2 > 0$. [1] This correlator was originally defined in [3] in terms of derivatives acting on the localised partition function of the $\mathcal{N} = 2^*$ SYM theory on $S^4$ [4], which can be expressed as a mass deformation of the $\mathcal{N} = 4$ theory. The $\mathcal{N} = 4$ integrated correlator results from the $m \to 0$ limit (where $m$ is the hypermultiplet mass). In this paper, we will consider an integrated correlator for $\mathcal{N} = 4$ SYM with any classical gauge group $G_N = SU(N), SO(2N), SO(2N+1), USp(2N)$, which is given by

$$\mathcal{C}_{G_N}(\tau,\bar{\tau}) = \frac{1}{4}\Delta_\tau \partial_m^2 \log Z_{G_N}(m,\tau,\bar{\tau})\Big|_{m=0}, \tag{1}$$

---

[1]The action of $SL(2,\mathbb{Z})$ is: $\tau \underset{SL(2,\mathbb{Z})}{\to} (a\tau + b)/(c\tau + d)$, where $a, b, c, d \in \mathbb{Z}$ and $ad - bc = 1$.

where $Z_{G_N}(m, \tau, \bar{\tau})$ is the partition function of $\mathcal{N} = 2^*$ SYM on $S^4$ with a gauge group $G_N$, $\mathcal{C}_{G_N}(\tau, \bar{\tau})$ denotes the integrated four-point correlator and $\Delta_\tau = \tau_2^2(\partial_{\tau_1}^2 + \partial_{\tau_2}^2)$ is the laplacian on the hyperbolic plane. The expression (1) was shown in [3] to define a four-point correlator integrated over the positions of the operators with a specific measure that has the following schematic form

$$\int \prod_{i=1}^4 dx_i \, \mu(x_1, \ldots, x_4) \langle \mathcal{O}_2(x_1) \ldots \mathcal{O}_2(x_4) \rangle, \tag{2}$$

where $\mathcal{O}_2(x)$ denotes the superconformal primary operator in the stress tensor supermultiplet, which is in the **20'** of the $SU(4)$ R-symmetry group and $\mu(x_1, \ldots, x_4)$ is a measure factor. The precise expression for (2) is discussed in [3] and later references. Some properties of the large-$N$ expansion of $\mathcal{C}_{SU(N)}(\tau, \bar{\tau})$ were considered in [5–7].[2]

A second integrated correlator of the form (2) but with a different integration measure was introduced in [7], and is proportional to $\partial_m^4 \log Z_{SU(N)}(m, \tau, \bar{\tau})\big|_{m=0}$. Some properties of its large-$N$ expansion were elucidated in [8] and more recently in [9]. We will not consider this integrated correlator in this paper.

## 1.1 The main results

In this paper we will consider the extension of the $SU(N)$ results of [1,2] to the other classical Lie groups, $SO(2N)$, $SO(2N+1)$, and $USp(2N)$. Some aspects of the perturbative expansions of the integrated correlators for these groups, and their large-$N$ expansions in the 't Hooft limit were considered in [10] starting from the localised partition function of $\mathcal{N} = 2^*$ SYM described in [4]. Our analysis will include the non-perturbative instanton contributions, leading to expressions for the integrated correlators for $\mathcal{N} = 4$ SYM with any classical gauge group that take the form of two-dimensional lattice sums[3]

$$\mathcal{C}_{G_N}(\tau, \bar{\tau}) = \sum_{(m,n)\in\mathbb{Z}^2} \int_0^\infty dt \left( B_{G_N}^1(t) e^{-t\pi \frac{|m+n\tau|^2}{\tau_2}} + B_{G_N}^2(t) e^{-t\pi \frac{|m+2n\tau|^2}{2\tau_2}} \right). \tag{3}$$

The rational functions $B_{G_N}^1(t)$ and $B_{G_N}^2(t)$ will be defined in detail later. Here we note that in the simply-laced cases, i.e. $SU(N)$ and $SO(2N)$, we have $B_{G_N}^2(t) = 0$ so that we may drop the superscript and denote $B_{G_N}^1(t) = B_{G_N}(t)$. In these cases the expression is manifestly invariant under $SL(2, \mathbb{Z})$, which is generated by the transformations $S$ and $T$ where $T : \tau \to \tau + 1$ and $S : \tau \to -1/\tau$. This was originally suggested by Montonen and Olive [11–13] following the observations by Goddard, Nuyts and Olive (GNO) concerning the relation between electric charge and magnetic monopole weight lattices in gauge field theories [14].

In the non simply-laced cases, i.e. $SO(2N+1)$ and $USp(2N)$, the expression (3) is invariant under $\Gamma_0(2) \subset SL(2, \mathbb{Z})$.[4] This is the group generated by $T$ and $\hat{S} T \hat{S}$, where $\hat{S} : \tau \to -1/(2\tau)$, $T : \tau \to \tau + 1$. The action of $\hat{S}$ does not leave (3) invariant but rather interchanges the two terms. However, we will see that

$$B_{SO(2N+1)}^1(t) = B_{USp(2N)}^2(t), \qquad B_{USp(2N)}^1(t) = B_{SO(2N+1)}^2(t), \tag{4}$$

so that $\hat{S}$ acts as a GNO (or Langlands) duality transformation [15–17], which relates $\mathcal{C}_{SO(2N+1)}$ with $\mathcal{C}_{USp(2N)}$. Since we are only concerned with correlation functions of local operators,

---

[2]In these references the correlator was denoted $\mathcal{G}_N(\tau, \bar{\tau})$.

[3]As we will clarify later, the $SO(3)$ case is an exception, and in that case the integrated correlator is $\mathcal{C}_{SO(3)}(\frac{\tau}{2}, \frac{\bar{\tau}}{2})$ (rather than $\mathcal{C}_{SO(3)}(\tau, \bar{\tau})$), which agrees with the result of supersymmetric localisation.

[4]An element $\gamma = \left(\begin{smallmatrix} a & b \\ c & d \end{smallmatrix}\right) \in SL(2, \mathbb{Z})$ belongs to the congruence subgroup $\Gamma_0(2)$ if $c = 0 \bmod 2$.

effectively GNO duality acts at the level of Lie algebras rather than Lie groups. The global versions of GNO duality are briefly reviewed in appendix A.

Detailed discussion of these results will be given in later sections but here we note the following general points:

• As in the $SU(N)$ case considered in [1,2] the functions $B^i_{G_N}(t)$ ($i = 1, 2$) satisfy inversion conditions

$$B^i_{G_N}(t) = t^{-1} B^i_{G_N}(t^{-1}),\tag{5}$$

and integration conditions

$$\int_0^\infty dt\, B_{SU(N)}(t) = \frac{N(N-1)}{8},$$
$$\int_0^\infty dt\, B_{SO(2N)}(t) = \int_0^\infty dt\, B^1_{SO(2N+1)}(t) = \int_0^\infty dt\, B^2_{USp(2N)}(t) = \frac{N(N-1)}{4},$$
$$\int_0^\infty dt\, B^2_{SO(2N+1)}(t) = \int_0^\infty dt\, B^1_{USp(2N)}(t) = \frac{N}{4},\tag{6}$$

as well as

$$\int_0^\infty \frac{dt}{\sqrt{t}}\, B^i_{G_N}(t) = 0.\tag{7}$$

• The integrated correlator (3) can be expressed as a formal expansion of the form

$$\mathcal{C}_{G_N}(\tau, \bar{\tau}) = -b_{G_N}(0) + \sum_{s=2}^\infty \left[ b^1_{G_N}(s) E(s; \tau, \bar{\tau}) + b^2_{G_N}(s) E(s; 2\tau, 2\bar{\tau}) \right],\tag{8}$$

where $E(s; \tau, \bar{\tau})$ is a non-holomorphic (or real analytic) Eisenstein series with $s \in \mathbb{N}$ (in our convention $E(0; \tau, \bar{\tau}) = -1$). The coefficients $b^1_{G_N}(s)$ and $b^2_{G_N}(s)$ are rational numbers that are determined by the expansion of $B^i_{G_N}(t)$ in the form

$$B^i_{G_N}(t) = \sum_{s=2}^\infty \frac{b^i_{G_N}(s)}{\Gamma(s)} t^{s-1}, \qquad i = 1, 2,\tag{9}$$

and $b_{G_N}(0) = b^1_{G_N}(0) + b^2_{G_N}(0)$ (since $b^2_{G_N}(s) = 0$ for $G_N = SU(N)$ and $SO(2N)$, in these cases we will drop the superscript and write $b^1_{G_N}(s) = b_{G_N}(s)$).

• It was pointed out in [9], in the $SU(N)$ case that the formal expression (8) can be written in a manifestly convergent manner using the conventional spectral representation for a modular invariant function. Similarly, (8) (which is a $\Gamma_0(2)$ invariant expression in the $SO(2N+1)$ and $USp(2N)$ cases) has the form,

$$\mathcal{C}_{G_N}(\tau, \bar{\tau}) = -2b_{G_N}(0) + \frac{1}{2\pi i} \int_{\frac{1}{2}-i\infty}^{\frac{1}{2}+i\infty} ds\, \frac{\pi(-1)^s}{\sin \pi s} \left[ b^1_{G_N}(s) E(s; \tau, \bar{\tau}) + b^2_{G_N}(s) E_s(s; 2\tau, 2\bar{\tau}) \right].\tag{10}$$

In [9] it was shown that the constant $-2b_{G_N}(0)$ is equal to the ensemble average $\langle \mathcal{C}_{G_N} \rangle$, i.e. the integral of $\mathcal{C}_{G_N}(\tau, \bar{\tau})$ over the $\mathcal{N} = 4$ conformal manifold, with respect to the Zamolodchikov metric.

• The expressions (3) and (8) transform covariantly under GNO duality. In the simply-laced cases, $SU(N)$ and $SO(2N)$, the coefficients $b^2_{G_N}(s)$ vanish. Since $E(s; \tau, \bar{\tau})$ is a modular function, the integrated correlators in these cases are invariant under $SL(2, \mathbb{Z})$.

• In the non simply-laced cases, $SO(2N+1)$ and $USp(2N)$, it follows from (4) and (9) that $\mathcal{C}_{G_N}(\tau, \bar{\tau})$ given by (3) is invariant under the $\Gamma_0(2)$ subgroup of $SL(2, \mathbb{Z})$ that is generated by the transformations $T$ and $\hat{S}T\hat{S}$. The action of $\hat{S}$ on $\mathcal{C}_{G_N}(\tau, \bar{\tau})$ effectively interchanges $b^1_{G_N}(s)$ and $b^2_{G_N}(s)$ since

$$E(s; \tau, \bar{\tau}) \underset{\hat{s}}{\to} E\left(s; -\frac{1}{2\tau}, -\frac{1}{2\bar{\tau}}\right) = E(s; 2\tau, 2\bar{\tau}). \tag{11}$$

This interchanges the integrated correlators for the $SO(2N+1)$ and $USp(2N)$ cases, and is interpreted as a GNO duality transformation.

• The integrated correlators also satisfy Laplace-difference equations that generalise the equation satisfied in the $SU(N)$ case in [1,2]. These take the schematic form

$$\Delta_\tau \mathcal{C}_{G_N} - 2c_{G_N}\left[\mathcal{C}_{G_{N+1}} - 2\mathcal{C}_{G_N} + \mathcal{C}_{G_{N-1}}\right]$$
$$+ d_{G_{N+1}}\mathcal{C}_{SU(2N-1)} + d_{G_N}\mathcal{C}_{SU(2N)} + d_{G_{N-1}}\mathcal{C}_{SU(2N+1)} = 0, \tag{12}$$

for $G_N = SO(2N), SO(2N+1), USp(2N)$, and where $c_{G_N}$ is the central charge. The precise values for the coefficients $d_{G_{N+1}}, d_{G_N}, d_{G_{N-1}}$ will be given later for each gauge group. These equations also display the anticipated covariance under GNO duality.

• The Laplace-difference equation for $SO(2N)$ is mapped into the Laplace-difference equation for $USp(2N)$ under the transformation $N \to -N$, together with $\tau \to -2\tau$. We will furthermore see that the perturbative expansions of the integrated correlators confirm the identification of $\mathcal{C}_{SO(-2N)}(-\tau, -\bar{\tau})$ and $\mathcal{C}_{USp(2N)}(2\tau, 2\bar{\tau})$.

• The large-$N$ expansion is naturally expressed as an expansion in inverse half-integer powers of the Ramond–Ramond (RR) five-form flux,

## 1.2  Outline

In section 2 we will present some properties of the integrated correlator, $\mathcal{C}_{G_N}$, defined in (1) in terms of derivatives of the partition function of $\mathcal{N} = 2^*$ SYM on $S^4$ in the $m \to 0$ limit. These results are based on methods outlined in appendix B, which includes a brief summary of the perturbative structure of integrated correlators given in [10], and an overview of instanton calculations based on the Nekrasov partition function [18] generalisied to arbitrary classical gauge groups [19,20]. The perturbation expansions for $\mathcal{C}_{G_N}(\tau_2)$ with finite $N$ are presented in section 2.1. The expansions for $\mathcal{C}_{SO(2N)}, \mathcal{C}_{SO(2N+1)}$ and $\mathcal{C}_{USp(2N)}$ generalise the expansion of $\mathcal{C}_{SU(N)}$ and display a number of interesting features, such as the equality of the $SO(2N)$ and $USp(-2N)$ integrated correlators when $g^2_{YM} \to -2g^2_{YM}$. Furthermore, when expressed in terms of appropriate expansion parameters all three of these integrated correlators have identical planar contributions (where the definition of 'planar' is dependent on the gauge group). Non-planar contributions begin at $O\left((g^2_{YM})^4\right)$. The instanton contributions to $\mathcal{C}_{G_N}$ are discussed in section 2.2 based on the formalism described in appendix B.3. The explicit form of these instanton contributions to $\mathcal{C}_{G_N}$ is difficult to extract from the localised partition function for general instanton number. However, we have determined the exact expressions for the one-instanton sector, and to a certain extent the two- and three-instanton sectors.

In section 3 we will demonstrate that the perturbative parts of the integrated correlators satisfy 'Laplace-difference' equations that have a form illustrated in (12), which imply powerful constraints on their structure. By studying various examples of these equations we are led in section 4 to conjecture that the fully non-perturbative expression for an integrated correlator $\mathcal{C}_{G_N}(\tau, \bar{\tau})$ can be expressed as the two-dimensional lattice sum in (3), which is formally equivalent to the infinite sum of non-holomorphic Eisenstein series of integer index in (8). These expressions transform in a manifestly covariant fashion under GNO duality. They also contain

an infinite number of Yang–Mills instanton contributions with precisely specified properties, which we will demonstrate agree with the instanton contributions to the localised correlators obtained in section 2.2. The arguments that motivate the Laplace-difference equations are presented in appendix C.

In section 5 we will consider the large-$N$ expansion of $\mathcal{C}_{G_N}(\tau, \bar{\tau})$ in various limits of the Yang–Mills coupling. In both the weakly-coupled and strongly-coupled 't Hooft limits considered in section 5.1 the instanton contributions are suppressed exponentially in $N$ and only the perturbative terms contribute. As we will show, if we introduce suitable expansion parameters the perturbative expansions for different gauge groups are closely related. The definitions of these parameters, which are generalisations of the parameters $N$ and $g_{YM}^2 N$ for the $SU(N)$ case, that are suited to the large-$N$ weak-coupling expansion are not generally the same as the parameters suited to the large-$N$ strong-coupling expansion. In section 5.2, we consider the large-$N$ limit with fixed $g_{YM}^2$, where the instanton contribution is crucial for exhibiting manifest invariance under GNO duality. The expressions for the integrated correlators, which are obtained by solving Laplace-difference equations, take their most compact form when expanded in inverse (half-integral) powers of Ramond–Ramond five-form flux $\tilde{N}_{G_N}$. The powers of $1/\tilde{N}_{G_N}$ correspond to powers of $\alpha'$ in the low energy expansion of the holographic dual string theory and beautifully match the expected string theory structure.

We will end in section 6 with a discussion of these results and of possible future directions.

## 2 Integrated correlators for general classical Lie groups

In this section we will determine properties of the perturbative and instantonic contributions to the integrated correlators based on supersymmetric localisation. The perturbative terms are contained in the zero Fourier mode with respect to $\tau_1$ whereas the non-perturbative terms correspond to the sum over instantons with instanton number $k \neq 0$. In other words, we can express the correlator as a Fourier series,

$$\mathcal{C}_{G_N}(\tau, \bar{\tau}) = \mathcal{C}_{G_N}^{(0)}(\tau_2) + \sum_{k=1}^{\infty} \left( e^{2\pi i k \tau} \mathcal{C}_{G_N}^{(k)}(\tau_2) + e^{-2\pi i k \bar{\tau}} \mathcal{C}_{G_N}^{(-k)}(\tau_2) \right), \tag{13}$$

where the $k = 0$ term is the perturbative contribution,

$$\mathcal{C}_{G_N}^{pert}(\tau_2) := \mathcal{C}_{G_N}^{(0)}(\tau_2), \tag{14}$$

and the $k \neq 0$ terms are the instanton and anti-instanton contributions,

$$\mathcal{C}_{G_N}^{inst}(\tau, \bar{\tau}) := \sum_{k=1}^{\infty} \left( e^{2\pi i k \tau} \mathcal{C}_{G_N}^{(k)}(\tau_2) + e^{-2\pi i k \bar{\tau}} \mathcal{C}_{G_N}^{(-k)}(\tau_2) \right). \tag{15}$$

Since the integrated correlator is real it follows that $\mathcal{C}_{G_N}^{(k)}(\tau_2) = \mathcal{C}_{G_N}^{(-k)}(\tau_2)$ so that $\mathcal{C}_{G_N}(\tau, \bar{\tau})$ contains equal contributions from instantons and anti-instantons.

### 2.1 Perturbative contribution

The perturbative sectors of the integrated correlators, $\mathcal{C}_{G_N}^{pert}$ derived from the localised partition function, were discussed in [10], where they were expressed in terms of generalised Laguerre polynomials as reviewed in appendix B.2. One of the primary interests in [10] was to use this perturbative data to determine terms in the large-$N$ expansion order by order in $1/N$ or, more precisely, order by order in the inverse central charges, $1/c_{G_N}$. However, here we will study

the perturbation expressions at finite $N$ in more detail, which will motivate the form of a set of Laplace-difference equations that generalises the analysis in [1,2] of the $SU(N)$ correlators, as well as the modular covariant expressions (3) that are well-defined for all values of $N$ and $\tau$. Further strong evidence for these expressions will be obtained from the evaluation of the instanton contributions in the next subsection.

Our starting point is the explicit result for the perturbative sector $\mathcal{C}_{G_N}^{pert}$ obtained in [10], and for convenience summarised in appendix B. The expansions of the expressions in (135)-(138) in powers of $g_{YM}^2$ can be organised in a striking manner by defining the expansion parameters, $a_{G_N}$, for each gauge group in the following manner[5]

$$a_{SU(N)} = \frac{N g_{YM}^2}{4\pi^2}, \qquad a_{SO(n)} = \frac{(n-2)g_{YM}^2}{4\pi^2}, \qquad a_{USp(n)} = \frac{(n+2)g_{YM}^2}{8\pi^2}, \tag{16}$$

where $n = 2N$ or $2N + 1$ for $SO(n)$, and $n = 2N$ for $USp(n)$.[6] We note that $a_{SU(N)}$ is the 't Hooft coupling of the $SU(N)$ theory (up to a factor $4\pi^2$), while $a_{SO(n)}$ and $a_{USp(n)}$ are the generalisations for $SO(n)$ and $USp(n)$ theory (see also [21]). [7]

The perturbative 't Hooft couplings defined in (16) can be rewritten in the compact form $a_G = h_G^\vee g_{YM}^2/(4\pi^2)$, with $h_G^\vee$ the dual Coxeter number for the group $G$. The appearance of the dual Coxeter number is quite natural in $\mathcal{N} = 4$ SYM when all the fields belong to the adjoint representation.

In terms of these parameters we find that the perturbative expansion of all the integrated correlators can be expressed in the following form,

$$\begin{aligned} \mathcal{C}_{G_N}^{pert}(\tau_2) = -4c_{G_N} \Bigg[ &\frac{3\,\zeta(3)a_{G_N}}{2} - \frac{75\,\zeta(5)a_{G_N}^2}{8} + \frac{735\,\zeta(7)a_{G_N}^3}{16} - \frac{6615\,\zeta(9)\left(1 + P_{G_N,1}\right)a_{G_N}^4}{32} \\ &+ \frac{114345\,\zeta(11)\left(1 + P_{G_N,2}\right)a_{G_N}^5}{128} - \frac{3864861\,\zeta(13)\left(1 + P_{G_N,3}\right)a_{G_N}^6}{1024} \\ &+ \frac{32207175\,\zeta(15)\left(1 + P_{G_N,4}\right)a_{G_N}^7}{2048} + \mathcal{O}(a_{G_N}^8) \Bigg], \end{aligned} \tag{17}$$

where $c_{G_N}$ is the conformal anomaly or central charge associated with $G_N$ and is given by

$$c_{SU(N)} = \frac{N^2 - 1}{4}, \qquad c_{SO(n)} = \frac{n(n-1)}{8}, \qquad c_{USp(n)} = \frac{n(n+1)}{8}. \tag{18}$$

We see that the first three perturbative contributions are universal and their dependence on $N$ is contained entirely within $c_{G_N}$ and $a_{G_N}$. Explicit "non-planar" factors, $P_{G_N,i}$, where $i = \ell - 3$ and $\ell$ is the loop number, first enter at four loops and the first few examples are listed below:

- $SU(N)$

$$\begin{aligned} P_{SU(N),1} &= \frac{2}{7N^2}, & P_{SU(N),2} &= \frac{1}{N^2}, \\ P_{SU(N),3} &= \frac{25N^2 + 4}{11N^4}, & P_{SU(N),4} &= \frac{605N^2 + 332}{143N^4}. \end{aligned} \tag{19}$$

---

[5]Note that the definition of $a_{G_N}$ differs from that in [10].

[6]The symbol $n$ is introduced is to unify the formulae for $SO(2N)$ and $SO(2N + 1)$, and to show the connection between $USp(2N)$ and $SO(2N)$ correlators.

[7]In the case of $SO(3)$, one needs to rescale $g_{YM} \to \sqrt{2}\,g_{YM}$ and define $a_{SO(3)} = g_{YM}^2/(2\pi^2)$ so that $a_{SO(3)} = a_{SU(2)} = a_{USp(2)}$. See also discussion below (138). Furthermore, one can see that $a_{SU(4)} = a_{SO(6)}$ and $a_{USp(4)} = a_{SO(5)}$, consistent with the isomorphic relations among these groups.

- $SO(n)$

$$P_{SO(n),1} = -\frac{n^2 - 14n + 32}{14(n-2)^3}, \qquad P_{SO(n),2} = -\frac{n^2 - 14n + 32}{8(n-2)^3},$$

$$P_{SO(n),3} = -\frac{12n^4 - 221n^3 + 1158n^2 - 2432n + 1856}{22(n-2)^5},$$

$$P_{SO(n),4} = -\frac{2\left(342n^5 - 7217n^4 - 48841n^3 - 153938n^2 + 239232n - 149920\right)}{715(n-2)^6}.$$

$$(20)$$

- $USp(n)$

$$P_{USp(n),1} = \frac{n^2 + 14n + 32}{14(n+2)^3}, \qquad P_{USp(n),2} = \frac{n^2 + 14n + 32}{8(n+2)^3},$$

$$P_{USp(n),3} = \frac{12n^4 + 221n^3 + 1158n^2 + 2432n + 1856}{22(n+2)^5},$$

$$P_{USp(n),4} = \frac{2\left(342n^5 + 7217n^4 + 48841n^3 + 153938n^2 + 239232n + 149920\right)}{715(n+2)^6}.$$

$$(21)$$

Some interesting features of these expansions are as follows.

- Whereas the genus expansion of $SU(N)$ gauge theory in powers of $1/N^2$ and $a_{SU(N)}$ [22] is well known, there seems to be no systematic analysis in the literature of the analogous expansions for $SO(n)$ and $USp(n)$ (although there are some limited results in [21]). We see from (16), (17), (20) and (21) that these expansions are purely in powers of $1/(n-2)$ and $1/(n+2)$, respectively. Indeed, if we define the parameters

$$N_{SU(N)} = N^2 = (h^\vee_{SU(N)})^2, \qquad N_{SO(n)} = n - 2 = h^\vee_{SO(n)}, \qquad N_{USp(n)} = n + 2 = 2h^\vee_{USp(n)},$$

$$(22)$$

the expansion (20) can be re-expressed in a form that generalises the topological expansion of the $SU(N)$ case, in which it takes the general form

$$\mathcal{C}_{G_N}(\tau, \bar\tau) \sim \mathcal{C}^{pert}_{G_N}(\tau_2) \sim c_{G_N} \sum_{g=0}^{\infty} (N_{G_N})^{-g} \mathcal{C}^{(g)}_{G_N}(a_{G_N}), \tag{23}$$

where the coefficients[8] $\mathcal{C}^{(g)}_{G_N}(a_{G_N})$ are power series, with rational coefficients, in the expansion parameter $a_{G_N}$ defined in (16). Following the terminology in the $SU(N)$ case, we will refer to terms with $g \geq 1$ as "non-planar" terms.

- A striking property of (17) is that the expression for the planar contribution $\mathcal{C}^{(0)}_{G_N}(a_{G_N})$ is the same for all the groups, and the non-planar contributions only enter at $\ell \geq 4$ loops, i.e. $\mathcal{C}^{(1)}_{G_N}(a_{G_N}) = O(a^4_{G_N})$. Such a property can be seen directly from the construction of perturbative loop integrands using the methods in [23,24], and will have important consequences when we consider the large-$N$ expansions. This property is only manifest with definition of the expansion parameters given in (16).

  Furthermore, the precise coefficients at each order of the perturbative expansion given in (17) can be verified using standard quantum field theory results. This calculation was described for the first two loops in [2] and for the planar terms up to order $O(a^4_{G_N})$

---

[8]The seemingly strange choice for $N_{SU(N)} = N^2$ is such that for the case of $SU(N)$ we obtain exactly the standard genus expansion of the form $c_{SU(N)} \sum_{g \geq 0} N^{-2g} \mathcal{C}^{(g)}_{SU(N)}(a_{SU(N)})$.

in [25] by explicitly performing the relevant higher-loop integrals. These results make use of the perturbative loop integrands constructed in [23, 24, 26, 27] and the precise expression for the integrated correlator (2) (see e.g. (2.3) of [2]).

- We should stress that the definition of the expansion parameters, $a_{G_N} = h^{\vee}_{G_N} g^2_{YM} /(4\pi^2)$ defined in (22), differ from the parameters that enter in the large-$N$ expansion in the holographic limit, which will be considered in section 5.1. In that case the parameters, which are denoted $\tilde{N}_{G_N}$, are defined in (97) in terms of the Ramond–Ramond five-form flux of an orientifold background. This is reviewed in appendix D. It is only if we use the expansion parameters defined in (22) that the weak-coupling perturbative expansion (17) has a finite number of non-planar terms, i.e. terms that are suppressed by powers of $1/N_{G_N}$, at fixed loop order $O(a^{\ell}_{G_N})$.

- The symmetry under the interchange $(N, g^2_{YM}) \leftrightarrow (-N, -g^2_{YM})$ is evident from the form of (17). For $SU(N)$, we have

$$c_{SU(N)} = c_{SU(-N)}, \qquad a_{SU(N)} = a_{SU(-N)}, \qquad P_{SU(N),i} = P_{SU(-N),i}, \qquad (24)$$

hence

$$\mathcal{C}^{pert}_{SU(N)}(g^2_{YM}) = \mathcal{C}^{pert}_{SU(-N)}(-g^2_{YM}), \qquad (25)$$

which reflect a relation between $SU(N)$ and $SU(-N)$. There are also relations between $SO(2N)$ and $USp(-2N)$ under $(N, g^2_{YM}) \leftrightarrow (-N, -2g^2_{YM})$

$$c_{SO(2N)} = c_{USp(-2N)}, \qquad a_{SO(2N)} = 2a_{USp(-2N)}, \qquad P_{SO(2N),i} = P_{USp(-2N),i}, \qquad (26)$$

which lead to

$$\mathcal{C}^{pert}_{SO(2N)}(g^2_{YM}) = \mathcal{C}^{pert}_{USp(-2N)}(-2g^2_{YM}). \qquad (27)$$

These relations have been further checked at higher orders. We will return to this point later in the discussion of the Laplace-difference equations.

## 2.2 Yang–Mills instanton sectors

In order to discuss the instanton contributions to $\mathcal{C}_{G_N}$ we will make use of the expressions shown in appendix B.3 for the contribution of instantons to the $\mathcal{N} = 2^*$ SYM partition function, $\hat{Z}^{inst}(m, a_i)$ that were obtained in [19, 20]. In particular, the full non-perturbative sector, presented in (15), can be computed from

$$\mathcal{C}^{inst}_{G_N}(\tau, \bar{\tau}) = \tau^2_2 \partial_\tau \partial_{\bar{\tau}} \partial^2_m Z^{inst}_{G_N}(m, \tau, \bar{\tau})\big|_{m \to 0}, \qquad (28)$$

with $Z^{inst}_{G_N}(m, \tau, \bar{\tau})$ the non-perturbative contribution to the localised $\mathcal{N} = 2^*$ partition function. As briefly reviewed in appendix B, $Z^{inst}_{G_N}$ can be obtained by a suitable matrix model integral over the variables $a_i$ of the Nekrasov partition function $\hat{Z}^{inst}_{G_N}(m, \tau, a_i)$. The $k$-instanton contribution to the Nekrasov partition function follows from the Fourier sum (139)

$$\hat{Z}^{inst}_{G_N}(m, \tau, a_i) = \sum_{k=1}^{\infty} e^{2\pi i k\tau} \hat{Z}^{(k)}_{G_N}(m, a_i) + c.c., \qquad (29)$$

where the complex conjugate, indicated by $c.c.$, contains the anti-instanton contribution.

The small-$m$ expansion of the $k$-instanton contribution for $SU(N)$ was well studied in [7] and led to the following compact expression,

$$\partial_m^2 \hat{Z}_{SU(N)}^{(k)}(m, a_i)\big|_{m=0} = \sum_{\substack{p,q>0 \\ pq=k}} \oint \frac{dz}{2\pi} \prod_{a=1}^{p} \prod_{b=1}^{q} \prod_{j=1}^{N} \frac{(z - a_j + i k_{a,b})^2}{(z - a_j + i k_{a,b})^2 + 1} \times \left[ \left( \frac{2}{p^2} + \frac{2}{q^2} \right) \right.$$

$$\left. + \sum_{j=1}^{N} \frac{i(q+p)(q-p)^2}{pq[z - a_j + i(p+q-1)][z - a_j + i(q-1)][z - a_j + i(p-1)]} \right], \tag{30}$$

where the integration contour $z$ is a counter-clockwise contour surrounding the poles at $z = a_j + i$ (with $j = 1, \ldots, N$) and $k_{a,b} = a + b - 2$.

In appendix B.3 we briefly summarise the results of [19,20] regarding the computation of the instantonic sectors via equivariant supersymmetric localisation for $\mathcal{N} = 4$ SYM with gauge groups $SO(2N)$, $SO(2N+1)$ and $USp(2N)$. Here we only present the results in the special case of relevance to us, in which the omega deformation parameters are set to $\epsilon_1 = \epsilon_2 = 1$, which amounts to localisation on $S^4$. We will only consider the complete expression in the single-instanton case ($k = 1$), and determine multiple-instanton contributions only for certain particular values of $N$. The general procedure is presented in appendix B.3, based on [19,20]. Here we will determine the explicit small-$m$ expansion of these results, which are relevant for the computation of the integrated correlators.

- $SO(2N)$:

  The one-instanton contribution for $SO(2N)$ is obtained by performing a one-dimensional contour integral using (140) and (141). The relevant poles are at $\phi_1 = a_j + \epsilon_+/2, -a_j + \epsilon_+/2, \epsilon_3/2, \epsilon_4/2$ [19]. Collecting all these residues, setting $\epsilon_1 = \epsilon_2 = 1$, and taking small-$m$ expansion, we find,

  $$\partial_m^2 \hat{Z}_{SO(2N)}^{(1)}(m, a_i)\big|_{m=0} = \sum_{j=1}^{N} \left( R_{a_j + \epsilon_+/2} + R_{-a_j + \epsilon_+/2} \right) + R_{\epsilon_3/2} + R_{\epsilon_4/2}, \tag{31}$$

  where $R_X$ is the result of taking residue at the pole at $\phi_1 = X$, and they are given by

  $$R_{\pm a_j + \epsilon_+/2} = \frac{2(\pm i a_j + 1)(\pm a_j + 2)}{(\pm 2 i a_j + 3)^2} \prod_{\ell \neq j} \frac{[(\pm i a_j + 1)^2 + a_\ell^2]^2}{[a_\ell^2 - a_j^2][(\pm i a_j + 2)^2 + a_\ell^2]},$$

  $$R_{\epsilon_3/2} + R_{\epsilon_4/2} = -\partial_m^2 \left[ \frac{m(m-3)}{32} \prod_{j=1}^{N} \frac{4 a_j^2 + (3m-1)^2}{4 a_j^2 + (m-3)^2} + (m \to -m) \right]\bigg|_{m=0}. \tag{32}$$

  In the final expression we have used the continuation $a_j \to i a_j$, which will also be used for the $SO(2N+1)$ and $USp(2N)$ cases considered below.

  Although we have not considered the general $k$-instanton expression for general $N$, we have evaluated special examples using the prescription for contour integrals that is discussed in appendix B.3. For example, the $k = 2$ contribution to the integrated correlator in the $SO(4)$ case has the form

  $$\partial_m^2 \hat{Z}_{SO(4)}^{(2)}(m, a_i)\big|_{m=0} = \frac{51}{16} - \frac{6}{(a_{12}^+)^2 + 9} - \frac{6}{a_{12}^2 + 9} + \frac{12}{[(a_{12}^+)^2 + 9]^2} + \frac{12}{[a_{12}^2 + 9]^2}, \tag{33}$$

  where $a_{ij} = a_i - a_j$ and $a_{ij}^+ = a_i + a_j$.

- $SO(2N+1)$:

  The computation for $SO(2N+1)$ is similar. In this case the one-instanton contribution for general $N$ is given by

  $$\partial_m^2 \hat{Z}_{SO(2N+1)}^{(1)}(m,a_i)\big|_{m=0} = \sum_{j=1}^{N}\left(R_{a_j+\epsilon_+/2} + R_{-a_j+\epsilon_+/2}\right) + R_{\epsilon_3/2} + R_{\epsilon_4/2}. \tag{34}$$

  Again $R_X$ is the result of taking residue at the pole at $\phi_1 = X$, and each takes the following form,

  $$R_{\pm a_j+\epsilon_+/2} = \frac{2(\pm i a_j + 1)^3}{\pm i a_j(\pm 2 i a_j + 3)^2}\prod_{\ell\neq j}\frac{\left[(\pm i a_j + 1)^2 + a_\ell^2\right]^2}{\left(a_\ell^2 - a_j^2\right)\left[(\pm i a_j + 2)^2 + a_\ell^2\right]},$$
  $$R_{\epsilon_3/2} + R_{\epsilon_4/2} = -\partial_m^2\left[\frac{m(3m-1)}{32}\prod_{j=1}^{N}\frac{4a_j^2 + (3m-1)^2}{4a_j^2 + (m-3)^2} + (m\to -m)\right]\bigg|_{m=0}. \tag{35}$$

  As a special example we have evaluated the $k=2$ contribution for the $SO(5)$ theory, which has the form

  $$\partial_m^2 \hat{Z}_{SO(5)}^{(2)}(m,a_i)\big|_{m=0} = \left(\frac{47}{32} - \frac{2}{a_1^2+4} + \frac{4}{\left(a_1^2+4\right)\left(a_2^2+4\right)}\right.$$
  $$\left. - \frac{6\left[2\left(a_1^4 - \left(a_2^2-21\right)a_1^2 + 7a_2^2 + 151\right)a_1^2 + 387\right]}{\left[a_1^4 - 2\left(a_2^2-9\right)a_1^2 + \left(a_2^2+9\right)^2\right]^2}\right) + (a_1\leftrightarrow a_2). \tag{36}$$

- $USp(2N)$:

  For the $USp(2N)$ group, the number of contour integrals is equal to $\lfloor\frac{k}{2}\rfloor$, with $k$ the instanton number. Therefore, no contour integral is involved in the one-instanton case. For this reason, the one-instanton contribution is given by the following compact expression,

  $$\partial_m^2 \hat{Z}_{USp(2N)}^{(1)}(m,a_i)\big|_{m=0} = \frac{1}{2}\prod_{j=1}^{N}\frac{a_j^2}{a_j^2+2}. \tag{37}$$

  When $k=2$ and $k=3$, the contour integrals are only one-dimensional, and are relatively easy to perform. For instance, the two- and three-instanton contributions for $USp(4)$ are found to be:

  $$\partial_m^2 \hat{Z}_{USp(4)}^{(2)}(m,a_i)\big|_{m=0} = \left(\frac{19}{16} - \frac{6}{2a_1^2+9} + \frac{12}{\left[2a_1^2+9\right]^2} - \frac{96}{\left(2a_1^2+9\right)\left[2a_2^2+9\right]^2}\right.$$
  $$\left. - \frac{8\left[4\left(a_2^2+3\right)a_1^4 + \left(32a_2^2+57\right)a_1^2 - 30\right]}{\left(2a_1^2+9\right)\left(2a_2^2+9\right)\left(a_{12}^2+8\right)\left[(a_{12}^+)^2+8\right]}\right) + (a_1\leftrightarrow a_2), \tag{38}$$

  and

  $$\partial_m^2 \hat{Z}_{USp(4)}^{(3)}(m,a_i)\big|_{m=0} = \frac{\left(a_2^4 + 20a_2^2 + 80\right)a_1^4 + 100\left(a_2^2+8\right)a_1^2 + 1024}{3\left[a_1^2+8\right]^2\left[a_2^2+8\right]^2} + (a_1\leftrightarrow a_2). \tag{39}$$

  We have also computed $\partial_m^2 \hat{Z}_{USp(2N)}^{(k)}(m,a_i)\big|_{m=0}$ for $USp(2N)$ for $k=2,3$, with $2\leq N\leq 5$. However, some of the expressions are somewhat lengthy and we will not show them explicitly here.

Once $\partial_m^2 \hat{Z}_{G_N}^{(k)}(m, a_i)\big|_{m=0}$ has been determined, it is straightforward to compute the matrix integrals using the expressions for expectation values given in (130), (132) and (134). We find the resulting instanton contributions to $\mathcal{C}_{G_N}^{inst}(\tau, \bar{\tau})$ agree precisely with the expected results based on the duality-covariant ansatz (3). We will return to this comparison in section 4 where we will discuss the ansatz and its motivation in more detail.

## 3 Laplace-difference equations

A striking property of the formulation of the $SU(N)$ integrated correlator in [1, 2] is that it satisfies a Laplace equation that relates it to the $SU(N-1)$ and $SU(N+1)$ correlators,

$$\Delta_\tau \mathcal{C}_{SU(N)}(\tau, \bar{\tau}) - 4c_{SU(N)}\Big[\mathcal{C}_{SU(N+1)}(\tau, \bar{\tau}) - 2\mathcal{C}_{SU(N)}(\tau, \bar{\tau}) + \mathcal{C}_{SU(N-1)}(\tau, \bar{\tau})\Big]$$
$$- (N+1)\mathcal{C}_{SU(N-1)}(\tau, \bar{\tau}) + (N-1)\mathcal{C}_{SU(N+1)}(\tau, \bar{\tau}) = 0. \tag{40}$$

This equation, which is reviewed in appendix C, has powerful consequences. Given the initial condition $\mathcal{C}_{SU(1)} = 0$, this equation easily determines the correlator for gauge group $SU(N)$ in terms of the correlator for gauge group $SU(2)$. Furthermore it gives a very simple iterative procedure for determining terms in the large-$N$ expansion of the correlator for gauge group $SU(N)$. We will now see how these statements generalise to any of the classical Lie groups.

Our procedure is to determine the Laplace-difference equations for general classical gauge groups by requiring consistency with the expressions determined in the previous section supplemented with the requirement of consistency with GNO duality. Using the perturbative results given in the section 2, we find that the integrated correlators obey equations of the form (12), in which the coefficients $d_{G_{N-1}}$, $d_{G_N}$ and $d_{G_{N+1}}$ are determined. Explicitly, we find the Laplace-difference equation for $SO(n)$ (with $n = 2N$ or $n = 2N+1$) is given by (more discussion of these equations is given in appendix C)

$$\Delta_\tau \mathcal{C}_{SO(n)}(\tau, \bar{\tau}) - 2c_{SO(n)}\Big[\mathcal{C}_{SO(n+2)}(\tau, \bar{\tau}) - 2\mathcal{C}_{SO(n)}(\tau, \bar{\tau}) + \mathcal{C}_{SO(n-2)}(\tau, \bar{\tau})\Big]$$
$$- n\mathcal{C}_{SU(n-1)}(\tau, \bar{\tau}) + (n-1)\mathcal{C}_{SU(n)}(\tau, \bar{\tau}) = 0. \tag{41}$$

The Laplace-difference equation for $USp(n)$ (with $n = 2N$) takes a very similar form,

$$\Delta_\tau \mathcal{C}_{USp(n)}(\tau, \bar{\tau}) - 2c_{USp(n)}\Big[\mathcal{C}_{USp(n+2)}(\tau, \bar{\tau}) - 2\mathcal{C}_{USp(n)}(\tau, \bar{\tau}) + \mathcal{C}_{USp(n-2)}(\tau, \bar{\tau})\Big]$$
$$+ n\mathcal{C}_{SU(n+1)}(2\tau, 2\bar{\tau}) - (n+1)\mathcal{C}_{SU(n)}(2\tau, 2\bar{\tau}) = 0. \tag{42}$$

Note that there is an important rescaling $(\tau, \bar{\tau}) \to (2\tau, 2\bar{\tau})$ in the $SU(N)$ correlators in the second line of (42).

**Lemma.** *Equations* (40) - (42) *can be solved iteratively to determine* $\mathcal{C}_{G_N}$ *for any classical Lie group* $G_N$, *once* $\mathcal{C}_{SU(2)}(\tau, \bar{\tau})$ *is given.*

*Proof.* The proof follows from identities satisfied by $\mathcal{C}_{G_N}(\tau, \bar{\tau})$ for small values of $N$.

- As discussed in [1, 2], the fact that the integrated correlator $\mathcal{C}_{SU(1)} = 0$ implies that the equation for $\mathcal{C}_{SU(N)}$ (40) can be solved for any $N$ in terms of $\mathcal{C}_{SU(2)}$ .

- The solutions for other groups follow by use of the identities: $\mathcal{C}_{USp(0)} = \mathcal{C}_{SO(0)} = \mathcal{C}_{SO(1)} = \mathcal{C}_{SO(2)} = 0$. Equation (41) with $n = 2$ and the fact that $\mathcal{C}_{SO(2)} = 0$ determine $\mathcal{C}_{SO(4)}$. Using $n = 2$ in (42) and $\mathcal{C}_{USp(2)}(\tau, \bar{\tau}) = \mathcal{C}_{SU(2)}(\tau, \bar{\tau})$ determines $\mathcal{C}_{USp(4)}$. Similarly, (41) with $n = 3$ and the fact that $\mathcal{C}_{SO(3)}(\tau, \bar{\tau}) = \mathcal{C}_{SU(2)}(2\tau, 2\bar{\tau})$ (remembering

that the localised $SO(3)$ correlator is actually $\mathcal{C}_{SO(3)}(\frac{\tau}{2}, \frac{\bar{\tau}}{2})$ as discussed earlier) determine $\mathcal{C}_{SO(5)}$.[9]

- Given the above initial conditions for small values of $N$, the solutions for arbitrary $N$ follow iteratively from the equations.

We can now consider a few examples of the solutions to the Laplace-difference equations using the procedure outlined above. This will help us to better understand the structures of the correlators and motivates a general ansatz for the integrated correlators, which we will discuss more detail in the next section. The general expression of $\mathcal{C}_{SU(N)}$, which may be obtained from (40), was given in [1,2]. Here we will consider the correlators in other gauge groups and use the general result of $\mathcal{C}_{SU(N)}$.

Let us begin with the correlators for $SO(2N)$. Using (41), it is straightforward to show that

$$\mathcal{C}_{SO(4)}(\tau, \bar{\tau}) = 2\mathcal{C}_{SU(2)}(\tau, \bar{\tau}), \qquad \mathcal{C}_{SO(6)}(\tau, \bar{\tau}) = \mathcal{C}_{SU(4)}(\tau, \bar{\tau}),$$

$$\mathcal{C}_{SO(8)}(\tau, \bar{\tau}) = -2\mathcal{C}_{SU(2)}(\tau, \bar{\tau}) + \frac{8}{3}\mathcal{C}_{SU(3)}(\tau, \bar{\tau}) - 2\mathcal{C}_{SU(4)}(\tau, \bar{\tau})$$
$$+ \frac{4}{5}\mathcal{C}_{SU(5)}(\tau, \bar{\tau}) + \frac{2}{3}\mathcal{C}_{SU(6)}(\tau, \bar{\tau}), \tag{43}$$

where we have used $\mathcal{C}_{SO(2)}(\tau, \bar{\tau}) = 0$. The expressions for $\mathcal{C}_{SO(4)}(\tau, \bar{\tau})$ and $\mathcal{C}_{SO(6)}(\tau, \bar{\tau})$ reflect the relations $SO(4) \cong SU(2) \times SU(2)$ and $SO(6) \cong SU(4)$, respectively. It is easy to see from the structure of the Laplace-difference equation that $\mathcal{C}_{SO(2N)}(\tau, \bar{\tau})$ can be expressed in terms of linear combination of $\mathcal{C}_{SU(m)}(\tau, \bar{\tau})$ with $m = 2, 3, \ldots, 2N-2$, as in the example of $\mathcal{C}_{SO(8)}(\tau, \bar{\tau})$ given above. As shown in [1,2], $\mathcal{C}_{SU(m)}(\tau, \bar{\tau})$ may be expressed as an infinite sum of the non-holomorphic Eisenstein series $E(s; \tau, \bar{\tau})$, or equivalently a two-dimensional lattice sum, hence the same is also true for $\mathcal{C}_{SO(2N)}(\tau, \bar{\tau})$, which we will discuss in more detail in the next section.

We now consider the integrated correlators in the $SO(2N+1)$ and $USp(2N)$ cases. We will see that the expressions for these correlators are related by GNO duality. To begin we will consider the first non-trivial correlators, $\mathcal{C}_{SO(5)}$ and $\mathcal{C}_{USp(4)}$. The Laplace-difference equations allow us to express the correlators in terms of the $SU(N)$ correlators,

$$\mathcal{C}_{SO(5)}(\tau, \bar{\tau}) = \left[-2\mathcal{C}_{SU(2)}(\tau, \bar{\tau}) + \frac{4}{3}\mathcal{C}_{SU(3)}(\tau, \bar{\tau})\right] + \left[-2\mathcal{C}_{SU(2)}(2\tau, 2\bar{\tau}) + \frac{4}{3}\mathcal{C}_{SU(3)}(2\tau, 2\bar{\tau})\right], \tag{44}$$

with an identical result for $\mathcal{C}_{USp(4)}(\tau, \bar{\tau})$, reflecting the fact that $USp(4) \cong SO(5)$. Using the results for $\mathcal{C}_{SU(N)}(\tau, \bar{\tau})$, we find $\mathcal{C}_{SO(5)}(\tau, \bar{\tau})$ (or equivalently $\mathcal{C}_{USp(4)}(\tau, \bar{\tau})$) can be also be expressed in terms of infinite sums of non-holomorphic Eisenstein series, but importantly involving both $E(s; \tau, \bar{\tau})$ and $E(s; 2\tau, 2\bar{\tau})$.

We will now consider $\mathcal{C}_{SO(7)}$ and $\mathcal{C}_{USp(6)}$, which will suggest the general structure of the integrated correlators and the GNO duality that relates $\mathcal{C}_{SO(2N+1)}$ and $\mathcal{C}_{USp(2N)}$. From (41) we find that $\mathcal{C}_{SO(7)}$ is given as a sum of $\mathcal{C}_{SU(N)}$ correlators of the form

$$\mathcal{C}_{SO(7)}(\tau, \bar{\tau}) = \left[\frac{8}{5}\mathcal{C}_{SU(2)}(\tau, \bar{\tau}) - \frac{12}{5}\mathcal{C}_{SU(3)}(\tau, \bar{\tau}) + \frac{3}{5}\mathcal{C}_{SU(4)}(\tau, \bar{\tau}) + \frac{4}{5}\mathcal{C}_{SU(5)}(\tau, \bar{\tau})\right]$$
$$+ \left[\frac{3}{5}\mathcal{C}_{SU(2)}(2\tau, 2\bar{\tau}) - \frac{12}{5}\mathcal{C}_{SU(3)}(2\tau, 2\bar{\tau}) + \frac{8}{5}\mathcal{C}_{SU(4)}(2\tau, 2\bar{\tau})\right], \tag{45}$$

---

[9]It should be emphasised that the initial conditions $\mathcal{C}_{SU(2)}(\tau, \bar{\tau}) = \mathcal{C}_{SO(3)}(2\tau, 2\bar{\tau}) = \mathcal{C}_{USp(2)}(\tau, \bar{\tau})$ are non-trivial properties. Using (135), (137), and (138), it is easy to check that their perturbative components are identical, and we have also confirmed that their non-perturbative terms agree.

and from (42) $\mathcal{C}_{USp(6)}$ we have

$$
\begin{aligned}
\mathcal{C}_{USp(6)}(\tau,\bar\tau) = & \left[\frac{8}{5}\mathcal{C}_{SU(2)}(2\tau,2\bar\tau) - \frac{12}{5}\mathcal{C}_{SU(3)}(2\tau,2\bar\tau) + \frac{3}{5}\mathcal{C}_{SU(4)}(2\tau,2\bar\tau) + \frac{4}{5}\mathcal{C}_{SU(5)}(2\tau,2\bar\tau)\right] \\
& + \left[\frac{3}{5}\mathcal{C}_{SU(2)}(\tau,\bar\tau) - \frac{12}{5}\mathcal{C}_{SU(3)}(\tau,\bar\tau) + \frac{8}{5}\mathcal{C}_{SU(4)}(\tau,\bar\tau)\right].
\end{aligned}
\tag{46}
$$

Since $\mathcal{C}_{SU(N)}(\tau,\bar\tau) = \mathcal{C}_{SU(N)}(-\frac{1}{\tau},-\frac{1}{\bar\tau})$ and $\mathcal{C}_{SU(N)}(2\tau,2\bar\tau) = \mathcal{C}_{SU(N)}(-\frac{1}{2\tau},-\frac{1}{2\bar\tau})$, it follows from (45) and (46) that under the transformation $\hat{S} : \tau \to -1/(2\tau)$, $\mathcal{C}_{SO(7)}(\tau,\bar\tau)$ transforms into $\mathcal{C}_{USp(6)}(\tau,\bar\tau)$. More generally, by induction, using the Laplace-difference equations (41) and (42), one can prove

$$
\mathcal{C}_{SO(2N+1)}(\tau,\bar\tau) = \mathcal{C}_{USp(2N)}\left(-\frac{1}{2\tau},-\frac{1}{2\bar\tau}\right),
\tag{47}
$$

which is the statement of GNO duality (recalling our previous comment that for $N=1$ the localised correlator equals $\mathcal{C}_{SO(3)}(\frac{\tau}{2},\frac{\bar\tau}{2})$, which also coincides with the modular invariant $\mathcal{C}_{SU(2)}(\tau,\bar\tau) = \mathcal{C}_{USp(2)}(\tau,\bar\tau)$). This property will be made manifest in the duality covariant ansatz of these correlators that will be proposed in the next section. It is also of note that these Laplace-difference equations are consistent with the dualities $\mathcal{C}_{SU(N)}(\tau,\bar\tau) = \mathcal{C}_{SU(-N)}(-\tau,-\bar\tau)$ and $\mathcal{C}_{SO(2N)}(\tau,\bar\tau) = \mathcal{C}_{USp(-2N)}(-\frac{\tau}{2},-\frac{\bar\tau}{2})$, which explicitly hold in perturbation theory, as we discussed earlier.

## 4 The duality covariant ansatz

In this section we will motivate the conjectured expression for $\mathcal{C}_{G_N}$ as the lattice sum (3). The argument for this expression will be based on the examples of solutions to the Laplace-difference equations presented in the previous section, which make it clear that the integrated correlators $\mathcal{C}_{SO(2N)}$, $\mathcal{C}_{SO(2N+1)}$ and $\mathcal{C}_{USp(2N)}$ can be written as linear combinations of $\mathcal{C}_{SU(m)}$ for certain values of $m$. The fact that $\mathcal{C}_{SU(N)}$ can be expressed (at least formally) as an infinite sum of non-holomorphic Eisenstein series [1, 2] suggests that $\mathcal{C}_{G_N}$ can also be expressed in terms of sums of Eisenstein series for any $G_N$. More precisely, we will find that $\mathcal{C}_{SO(2N)}$ is given by an infinite sum of $E(s;\tau,\bar\tau)$, whereas $\mathcal{C}_{SO(2N+1)}$ and $\mathcal{C}_{USp(2N)}$ involve both $E(s;\tau,\bar\tau)$ and $E(s;2\tau,2\bar\tau)$.

### 4.1 Review of $\mathcal{C}_{SU(N)}$

In [1,2] it was argued that the integrated correlator of $SU(N)$ theory can formally be expressed as an infinite sum of non-holomorphic Eisenstein series,

$$
\mathcal{C}_{SU(N)}(\tau,\bar\tau) = \frac{N(N-1)}{8} + \sum_{s=2}^{\infty} b_{SU(N)}(s)E(s;\tau,\bar\tau),
\tag{48}
$$

where the coefficients $b_{SU(N)}(s)$ are defined in terms of $B_{SU(N)}(t)$ by (9) and we have used $b_{SU(N)}(0) = -N(N-1)/8$.

In our normalisation, a non-holomorphic Eisenstein series is defined by

$$
E(s;\tau,\bar\tau) = \sum_{(m,n)\neq(0,0)} \frac{1}{\pi^s} \frac{\tau_2^s}{|m+n\tau|^{2s}},
\tag{49}
$$

which has the Fourier series expansion

$$E(s; \tau, \bar{\tau}) = \frac{2\zeta(2s)}{\pi^s} \tau_2^s + \frac{2\sqrt{\pi}\,\Gamma(s-\frac{1}{2})\zeta(2s-1)}{\pi^s \Gamma(s)} \tau_2^{1-s} \tag{50}$$

$$+ \sum_{\substack{k=-\infty \\ k \neq 0}}^{\infty} e^{2\pi i k \tau_1} \frac{4\sqrt{\tau_2}}{\Gamma(s)} |k|^{s-\frac{1}{2}} \sigma_{1-2s}(|k|) K_{s-\frac{1}{2}}(2\pi |k| \tau_2),$$

with $K_s$ a modified Bessel function of second kind and $\sigma_s(k) = \sum_{d|k} d^s$ a divisor function. The expression (48) is formal since it is not a convergent series, but it can be defined in a convergent manner by using the integral representation for the Eisenstein series

$$E(s; \tau, \bar{\tau}) = \sum_{(m,n) \neq (0,0)} \int_0^\infty e^{-t\pi \frac{|m+n\tau|^2}{\tau_2}} \frac{t^{s-1}}{\Gamma(s)} dt. \tag{51}$$

Substituting in (48) and using (6), gives a well-defined two-dimensional lattice sum expression [1,2],

$$\mathcal{C}_{SU(N)}(\tau, \bar{\tau}) = \sum_{(m,n) \in \mathbb{Z}^2} \int_0^\infty e^{-t\pi \frac{|m+n\tau|^2}{\tau_2}} B_{SU(N)}(t) dt, \tag{52}$$

where $B_{SU(N)}(t)$ is a rational function,

$$B_{SU(N)}(t) = \sum_{s=2}^\infty b_{SU(N)}(s) \frac{t^{s-1}}{\Gamma(s)} = \frac{\mathcal{Q}_{SU(N)}(t)}{(t+1)^{2N+1}}, \tag{53}$$

and $\mathcal{Q}_{SU(N)}(t)$ is a polynomial of degree $(2N-1)$ that takes the form

$$\mathcal{Q}_{SU(N)}(t) = -\frac{1}{4}N(N-1)(1-t)^{N-1}(1+t)^{N+1}$$

$$\left\{ (3 + (8N + 3t - 6)t) P_N^{(1,-2)}\left(\frac{1+t^2}{1-t^2}\right) + \frac{1}{1+t}\left(3t^2 - 8Nt - 3\right) P_N^{(1,-1)}\left(\frac{1+t^2}{1-t^2}\right) \right\}, \tag{54}$$

with $P_N^{(\alpha,\beta)}(z)$ being a Jacobi polynomial. It is notable that $B_{SU(-N)}(t) = B_{SU(N)}(-t)$ which is directly connected to the relation $\mathcal{C}_{SU(N)}(\tau, \bar{\tau}) = \mathcal{C}_{SU(-N)}(-\tau, -\bar{\tau})$.

A key feature of the function $B_{SU(N)}(t)$ in the representation of $\mathcal{C}_{SU(N)}$ in (52) is the inversion symmetry $B_{SU(N)}(t) = t^{-1}B_{SU(N)}(t^{-1})$. This property leads to particular relationships between the coefficients $b_{SU(N)}(s)$ in (9) that have important consequences. In particular, consider the zero Fourier mode of (48) (the perturbative sector), $\mathcal{C}_{SU(N)}^{pert}(\tau_2)$, which is the sum of infinitely many zero modes of Eisenstein series. From (50), we see that this results in the sum of two infinite series:

$$\mathcal{C}_{SU(N)}^{pert}(\tau_2) = \mathcal{C}_{SU(N)}^{(i)}(\tau_2) + \mathcal{C}_{SU(N)}^{(ii)}(\tau_2), \tag{55}$$

where $\mathcal{C}_{SU(N)}^{(i)}(\tau_2)$ denotes the sum of $\tau_2^{1-s}$ terms, which is asymptotic and gives a well-defined perturbative series for small $g_{YM}^2 = 4\pi/\tau_2$, and $\mathcal{C}_{SU(N)}^{(ii)}(\tau_2)$, which is the sum of $\tau_2^s$ terms, is divergent term by term as $g_{YM}^2 \to 0$. However, as presented in more detail in [2], the latter series can be Borel resummed and the result is

$$\mathcal{C}_{SU(N)}^{(ii)}(\tau_2) = \mathcal{C}_{SU(N)}^{(i)}(\tau_2) = \frac{1}{2}\mathcal{C}_{SU(N)}^{pert}(\tau_2), \tag{56}$$

we stress that the lattice sum representation (52) is a well-defined function for all values of $\tau$ in the upper-half plane and for all values of $N \geq 0$, while the need for Borel resummation only arises when it is expanded in perturbation theory.

The coefficients $b_{SU(N)}(s)$ in the ansatz (48) or, equivalently, the rational function $B_{SU(N)}(t)$, are uniquely determined by matching $\mathcal{C}^{pert}_{SU(N)}(\tau_2)$ with the perturbative terms determined by localisation. In other words, $B_{SU(N)}(t)$ can be determined by matching with the perturbative contributions given in (135), in a manner that we will now describe.

Let us begin by considering the perturbative contribution arising from the lattice-sum representation. Assume that $B_{SU(N)}(t)$ is given by a convergent expansion

$$B_{SU(N)}(t) = \sum_{s=1}^{\infty} \alpha(s) \, t^s \,. \tag{57}$$

Substituting this expression in (52) and computing the perturbative terms (in terms of the representation (48) we are using the fact that the $\tau_2^s$ terms sum to give the same contribution as the $\tau_2^{1-s}$ terms, as discussed above), one finds that the perturbative contribution is given by the asymptotic formal power series

$$\mathcal{C}^{pert}_{SU(N)}(y) \sim \sum_{s=1}^{\infty} \alpha(s) \frac{4 \Gamma\left(s + \frac{1}{2}\right) \zeta(2s+1)}{\sqrt{\pi}} y^{-s} \,, \tag{58}$$

with $y = \pi\tau_2 = 4\pi^2/g_{YM}^2$. We will now compare this result with the perturbative terms obtained from (135). For convenience we will denote the perturbative contribution to $\mathcal{C}_{SU(N)}(\tau, \bar{\tau})$ in (135) as

$$\mathcal{C}^{pert}_{SU(N)}(y) = \int_0^{\infty} \frac{d\omega}{\sinh^2 \omega} \omega \, y^2 \partial_y^2 K(\omega^2/y) \,, \tag{59}$$

and the integrand has the following convergent power series expansion

$$\omega \, y^2 \partial_y^2 K(\omega^2/y) = \sum_{s=1}^{\infty} \beta(s) \, \omega^{2s+1} y^{-s} \,. \tag{60}$$

Using the integral identity

$$\int_0^{\infty} d\omega \frac{\omega^{m+1}}{\sinh^2 \omega} = 2^{-m} \Gamma(m+2) \zeta(m+1) \,, \tag{61}$$

valid for $m \geq 1$, we obtain the perturbative contribution from (59), which is given by the asymptotic power series

$$\mathcal{C}^{pert}_{SU(N)}(y) \sim \sum_{s=1}^{\infty} \beta(s) \frac{\Gamma(2s+2) \zeta(2s+1)}{2^{2s}} y^{-s} \,. \tag{62}$$

By equating (58) and (62), we find the following relation between $\alpha(s)$ and $\beta(s)$

$$\beta(s) = \frac{4 \, \alpha(s)}{(2s+1)\Gamma(s+1)} \,. \tag{63}$$

Therefore knowing $y^2 \partial_y^2 K(\omega^2/y)$ allows us to determine $B_{SU(N)}(t)$. Explicitly, from (63), we find the following simple relationship[10]

$$B_{SU(N)}(t) = \frac{1}{4} \int_0^{\infty} dr \, e^{-r} \, \partial_{\omega} \left[ \omega \, y^2 \partial_y^2 K(\omega^2/y) \right] \Big|_{y=1, \omega=\sqrt{rt}} \,. \tag{64}$$

---

[10]Note that this procedure is closely related to the $SL(2, \mathbb{Z})$ Borel transform introduced in [9].

Using the expression for $K(\omega^2/y)$ given in (135), and after a suitable change of variables and integration by parts, the above expression can be recast into the following simpler form,

$$B_{SU(N)}(t) = -t\int_0^\infty dx\, e^{-xt}\tilde{B}_{SU(N)}(x),\tag{65}$$

where the integrand $\tilde{B}_{SU(N)}(x)$ is directly related to the perturbative result given in (135),

$$\tilde{B}_{SU(N)}(x) = \frac{x^{\frac{3}{2}}}{4}\partial_x\left\{x^{\frac{3}{2}}\partial_x\left[e^{-x}\sum_{i,j=1}^N\left(L_{i-1}(x)L_{j-1}(x) - (-1)^{i-j}L_{i-1}^{j-i}(x)L_{j-1}^{i-j}(x)\right)\right]\right\}.\tag{66}$$

Although proving that (65) is equivalent to (53) for arbitrary $N$ is rather non-trivial, it is straightforward to check explicitly the equivalence of these two expressions for any given $N$. We also note that the above derivation is general and not restricted to the $SU(N)$ case, therefore we will apply the same arguments for other gauge groups in the next subsection.

## 4.2 Exact expressions for $\mathcal{C}_{SO(2N)}, \mathcal{C}_{SO(2N+1)}$ and $\mathcal{C}_{USp(2N)}$

This discussion generalises to the other classical groups. As described in the previous section, the study of Laplace-difference equations makes it clear that the integrated correlators $\mathcal{C}_{SO(2N)}$, $\mathcal{C}_{SO(2N+1)}$ and $\mathcal{C}_{USp(2N)}$ can all be expressed as sums of Eisenstein series, as in the case of $SU(N)$. More precisely, the analysis of Laplace-difference equations suggests the following ansatz for the integrated correlator for each gauge group

$$\mathcal{C}_{SO(2N)}(\tau,\bar{\tau}) = \frac{N(N-1)}{4} + \sum_{s=2}^\infty b_{SO(2N)}(s)E(s;\tau,\bar{\tau}),\tag{67}$$

and[11]

$$\mathcal{C}_{SO(2N+1)}(\tau,\bar{\tau}) = \frac{N^2}{4} + \sum_{s=2}^\infty\left(b_{SO(2N+1)}^1(s)E(s;\tau,\bar{\tau}) + b_{SO(2N+1)}^2(s)E(s;2\tau,2\bar{\tau})\right),$$

$$\mathcal{C}_{USp(2N)}(\tau,\bar{\tau}) = \frac{N^2}{4} + \sum_{s=2}^\infty\left(b_{USp(2N)}^1(s)E(s;\tau,\bar{\tau}) + b_{USp(2N)}^2(s)E(s;2\tau,2\bar{\tau})\right),\tag{68}$$

and (47) implies

$$b_{SO(2N+1)}^1(s) = b_{USp(2N)}^2(s),\qquad b_{SO(2N+1)}^2(s) = b_{USp(2N)}^1(s),\tag{69}$$

since $\hat{S}$ exchanges $E(s;\tau,\bar{\tau})$ with $E(s;2\tau,2\bar{\tau})$. Similarly to $SU(N)$, for the constant term we have used the results $b_{SO(2N)}(0) = -N(N-1)/4$ and $b_{SO(2N+1)}(0) = b_{USp(2N)}(0) = -N^2/4$.

As in the case of $\mathcal{C}_{SU(N)}(\tau,\bar{\tau})$, these formal expressions are well-defined upon using the lattice sum representation of $E(s;\tau,\bar{\tau})$ in (51), which, using (6), leads to

$$\mathcal{C}_{SO(2N)}(\tau,\bar{\tau}) = \sum_{(m,n)\in\mathbb{Z}^2}\int_0^\infty e^{-t\pi\frac{|m+n\tau|^2}{\tau_2}}B_{SO(2N)}(t)dt,\tag{70}$$

where

$$B_{SO(2N)}(t) = \sum_{s=2}^\infty b_{SO(2N)}(s)\frac{t^{s-1}}{\Gamma(s)}.\tag{71}$$

---

[11]In the case of $\mathcal{C}_{SO(3)}$, $b_{SO(3)}^1(s) = 0$ and $b_{SO(3)}^2(s) = b_{SU(2)}(s)$, and to retrieve the localised correlator we should rescale $(\tau,\bar{\tau}) \to (\frac{\tau}{2},\frac{\bar{\tau}}{2})$, so that $\mathcal{C}_{SO(3)}(\frac{\tau}{2},\frac{\bar{\tau}}{2}) = \mathcal{C}_{SU(2)}(\tau,\bar{\tau})$.

Similarly, for $SO(2N+1)$ and $USp(2N)$, again using (6), we have

$$\mathcal{C}_{SO(2N+1)}(\tau,\bar{\tau}) = \sum_{(m,n)\in\mathbb{Z}^2} \int_0^\infty dt \left( B^1_{SO(2N+1)}(t) e^{-t\pi\frac{|m+n\tau|^2}{\tau_2}} + B^2_{SO(2N+1)}(t) e^{-t\pi\frac{|m+2n\tau|^2}{2\tau_2}} \right), \quad (72)$$

and

$$\mathcal{C}_{USp(2N)}(\tau,\bar{\tau}) = \sum_{(m,n)\in\mathbb{Z}^2} \int_0^\infty dt \left( B^1_{USp(2N)}(t) e^{-t\pi\frac{|m+n\tau|^2}{\tau_2}} + B^2_{USp(2N)}(t) e^{-t\pi\frac{|m+2n\tau|^2}{2\tau_2}} \right), \quad (73)$$

with $B^1_{SO(2N+1)}(t) = B^2_{USp(2N)}(t)$ and $B^2_{SO(2N+1)}(t) = B^1_{USp(2N)}(t)$, reflecting GNO duality.

The coefficients $b^i_{G_N}(s)$, or equivalently the rational functions $B^i_{G_N}(t)$, can again be determined by directly comparing the ansatz with the perturbative results. They can also be fixed using the Laplace-difference equations together with the expression for $\mathcal{C}_{SU(N)}(\tau,\bar{\tau})$. Either way, we find in the $SO(2N)$ case,

$$B_{SO(2N)}(t) = \frac{\mathcal{Q}_{SO(2N)}(t)}{(t+1)^{4N-3}}, \quad (74)$$

where $\mathcal{Q}_{SO(2N)}(t)$ is a palindromic polynomial of degree-$(4N-5)$. The following are specific examples,

$$\begin{aligned}
\mathcal{Q}_{SO(4)}(t) &= 2\mathcal{Q}_{SU(2)}(t) = 3t(3t^2 - 10t + 3), \\
\mathcal{Q}_{SO(6)}(t) &= \mathcal{Q}_{SU(4)}(t) = 15t\left(3t^6 - 23t^5 + 50t^4 - 72t^3 + 50t^2 - 23t + 3\right), \\
\mathcal{Q}_{SO(8)}(t) &= 126t\left(t^{10} - 12t^9 + 47t^8 - 122t^7 + 167t^6 - 182t^5 \right. \\
&\qquad \left. + 167t^4 - 122t^3 + 47t^2 - 12t + 1\right).
\end{aligned} \quad (75)$$

Just as in the $SU(N)$ case, since the coefficients $b_{SO(2N)}(s)$ are uniquely determined by the perturbation theory results, $B_{SO(2N)}(t)$ is related to the perturbative expression in terms of Laguerre polynomials (136). This allows us to obtain $B_{SO(2N)}(t)$ for arbitrary $N$ (65) by an analysis analogous to that used for $\mathcal{C}_{SU(N)}$, we find,

$$B_{SO(2N)}(t) = -t\int_0^\infty dx\, e^{-xt}\, \tilde{B}_{SO(2N)}(x), \quad (76)$$

with

$$\tilde{B}_{SO(2N)}(x) = \frac{x^{\frac{3}{2}}}{2}\partial_x\left\{x^{\frac{3}{2}}\partial_x\left[e^{-x}\sum_{i,j=1}^N \left(L_{2(i-1)}(x)L_{2(j-1)}(x) - L^{2(j-i)}_{2(i-1)}(x)L^{2(i-j)}_{2(j-1)}(x)\right)\right]\right\}. \quad (77)$$

One can easily verify that (76) reproduces the examples given in (75).

Similarly, for $SO(2N+1)$ (or equivalently $USp(2N)$), we find

$$B^1_{SO(2N+1)}(t) = B^2_{USp(2N)}(t) = \frac{\mathcal{Q}^1_{SO(2N+1)}(t)}{(t+1)^{4N-1}}, \quad (78)$$

$$B^2_{SO(2N+1)}(t) = B^1_{USp(2N)}(t) = \frac{\mathcal{Q}^2_{SO(2N+1)}(t)}{(t+1)^{2N+3}}, \quad (79)$$

where $\mathcal{Q}^1_{SO(2N+1)}(t)$ and $\mathcal{Q}^2_{SO(2N+1)}(t)$ are degree-$(4N-3)$ and degree-$(2N+1)$ palindromic polynomials, respectively. For $N = 1$, as previously mentioned, we have $\mathcal{C}_{SO(3)}(\tau,\bar{\tau}) = \mathcal{C}_{SU(2)}(2\tau,2\bar{\tau})$ hence, using (53), we deduce

$$B^1_{SO(3)}(t) = 0, \qquad B^2_{SO(3)}(t) = B_{SU(2)}(t) = \frac{3t(3t^2 - 10t + 3)}{2(t+1)^5}, \quad (80)$$

consistent with generic expectations (6).

It turns out that it is relatively simple to determine $\mathcal{Q}^2_{SO(2N+1)}(t)$. After examining many examples we find a simple expression

$$\mathcal{Q}^2_{SO(2N+1)}(t) = \frac{N}{2}(2N+1)t(t-1)^{2N-2}\left(3t^2 - (8N+2)t + 3\right), \tag{81}$$

consistent with $\mathcal{Q}^2_{SO(3)}(t) = \mathcal{Q}_{SU(2)}(t)$. Although a general formula for $\mathcal{Q}^1_{SO(2N+1)}(t)$ is harder to obtain, nevertheless one may compute it in principle for any $N$ either by use of the Laplace-difference equation or from the perturbative results. The following are two examples of $\mathcal{Q}^1_{SO(2N+1)}(t)$,

$$\begin{aligned}
\mathcal{Q}^1_{SO(5)}(t) &= \mathcal{Q}^2_{SO(5)}(t) = \mathcal{Q}^1_{USp(4)}(t) = \mathcal{Q}^2_{USp(4)}(t) = 15(t-1)^2 t\left(t^2 - 6t + 1\right), \\
\mathcal{Q}^1_{SO(7)}(t) &= 21t\left(3t^8 - 37t^7 + 123t^6 - 207t^5 + 220t^4 - 207t^3 + 123t^2 - 37t + 3\right).
\end{aligned} \tag{82}$$

As in the $SU(N)$ and $SO(2N)$ cases, the functions $B^i_{SO(2N+1)}(t)$ can be obtained from the perturbative expression in terms of sums of Laguerre polynomials given in (137). The term linear in Laguerre polynomials in (137) gives the simpler function, $B^2_{SO(2N+1)}(t)$,

$$B^2_{SO(2N+1)}(t) = -t\int_0^\infty dx\, e^{-xt}\, \tilde{B}^2_{SO(2N+1)}(x), \tag{83}$$

with[12]

$$\tilde{B}^2_{SO(2N+1)}(x) = \frac{x^{\frac{3}{2}}}{2}\partial_x\left[x^{\frac{3}{2}}\partial_x\left(e^{-x}\sum_{i=1}^{N}L_{2i-1}(2x)\right)\right], \tag{84}$$

and the term quadratic in Laguerre polynomials leads to $B^1_{SO(2N+1)}(t)$,

$$B^1_{SO(2N+1)}(t) = -t\int_0^\infty dx\, e^{-xt}\, \tilde{B}^1_{SO(2N+1)}(x), \tag{85}$$

with

$$\tilde{B}^1_{SO(2N+1)}(t) = \frac{x^{\frac{3}{2}}}{2}\partial_x\left\{x^{\frac{3}{2}}\partial_x\left[e^{-x}\sum_{i,j=1}^{N}\left(L_{2i-1}(x)L_{2j-1}(x) - L^{2(j-i)}_{2i-1}(x)L^{2(i-j)}_{2j-1}(x)\right)\right]\right\}. \tag{86}$$

Again, one can verify that (83) and (85) are in agreement with the expressions given in (81) and (82), respectively. In particular for $N = 1$ we can easily see that $B^1_{SO(3)}(t) = 0$ from (85) and $B^2_{SO(3)}(t) = B_{SU(2)}(t)$ from (83).

As described in the introduction, the functions $B^i_{G_N}(t)$ obey the inversion and integration conditions,

$$B^i_{G_N}(t) = t^{-1}B^i_{G_N}(t^{-1}), \qquad \int_0^\infty \frac{dt}{\sqrt{t}}B^i_{G_N}(t) = 0, \tag{87}$$

as well as the other integral conditions presented in (6), which we have checked for many different values of $N$. As explained in [9], both of these conditions are closely related to modularity of the corresponding lattice sum integrals.[13]

---

[12] The reason $L_{2i-1}(2x)$ (rather than $L_{2i-1}(x)$) arises in the definition of $\tilde{B}^2_{SO(2N+1)}(x)$ is because $B^2_{SO(2N+1)}(t)$ is associated with $E(s; 2\tau, 2\bar{\tau})$.

[13] We would like to thank Scott Collier and Eric Perlmutter for clarifications on this issue.

Finally, it would be interesting to obtain expressions for $B^i_{G_N}(t)$ as explicit functions of $N$ for general gauge group $G_N$, analogous to that of $SU(N)$ given in (54). Such expressions would allow us to perform non-perturbative checks of the relation $\mathcal{C}_{SO(2N)}(\tau, \bar{\tau}) = \mathcal{C}_{USp(-2N)}(-\frac{\tau}{2}, -\frac{\bar{\tau}}{2})$, which we have shown is a property of the perturbative expansion. Although it is difficult to perform the continuation $N \rightarrow -N$ using the expressions given in (77) and (86), we saw earlier that the Laplace-difference equations are perfectly consistent with this relation.

## 4.3 Non-perturbative checks

The coefficients $b^i_{G_N}(s)$, or equivalently $B^i_{G_N}(t)$, were designed to reproduce the perturbative expressions of the integrated correlators that are determined by localisation. It is important to verify that the exact expressions given by (3) also give rise to correct non-perturbative instanton contributions. We have already shown, using Laplace-difference equations, that $\mathcal{C}_{SO(4)}(\tau, \bar{\tau}) = 2\mathcal{C}_{SU(2)}(\tau, \bar{\tau})$ and $\mathcal{C}_{SO(6)}(\tau, \bar{\tau}) = \mathcal{C}_{SU(4)}(\tau, \bar{\tau})$ for any $\tau$, as expected. So here we will consider more general examples. For the one instanton contributions to $SO(n)$ correlators, we find:

$$\mathcal{C}^{(1)}_{SO(5)}(\tau, \bar{\tau}) = \mathcal{C}^{(1)}_{USp(4)}(\tau, \bar{\tau}) = e^{2\pi i \tau} 20 \Big[ y^2 (8y + 5) \tag{88}$$
$$- \frac{\sqrt{\pi}}{4} e^{4y} y^{3/2} \big( 64y^2 + 48y + 3 \big) \operatorname{erfc}(2\sqrt{y}) \Big],$$

$$\mathcal{C}^{(1)}_{SO(7)}(\tau, \bar{\tau}) = e^{2\pi i \tau} \frac{21}{32} \Big[ y^2 \big( 512y^3 + 2496y^2 + 2824y + 707 \big) \tag{89}$$
$$- \frac{\sqrt{\pi}}{4} e^{4y} y^{3/2} \big( 4096y^4 + 20480y^3 + 24960y^2 + 7936y + 317 \big) \operatorname{erfc}(2\sqrt{y}) \Big],$$

$$\mathcal{C}^{(1)}_{SO(8)}(\tau, \bar{\tau}) = e^{2\pi i \tau} \frac{7}{1024} \Big[ y^2 \big( 45056y^4 + 358400y^3 + 805632y^2 + 630336y + 136173 \big)$$
$$- \frac{\sqrt{\pi}}{4} e^{4y} y^{3/2} \big( 360448y^5 + 2912256y^4 + 6792192y^3 + 5765760y^2 \tag{90}$$
$$+ 1567800y + 60435 \big) \operatorname{erfc}(2\sqrt{y}) \Big],$$

where $y = \pi \tau_2 = 4\pi^2 / g^2_{YM}$. We have verified that all these results, as well as those with higher $N$, match precisely the one-instanton computation from localisation given in section 2.2.

Turning to $USp(2N)$ we first recall from (88) that $\mathcal{C}_{USp(4)}(\tau, \bar{\tau}) = \mathcal{C}_{SO(5)}(\tau, \bar{\tau})$. For higher values of $N$ we have, for example,

$$\mathcal{C}^{(1)}_{USp(6)}(\tau, \bar{\tau}) = e^{2\pi i \tau} 7 \Big[ y^2 (8y + 3)(8y + 11)$$
$$- \frac{\sqrt{\pi}}{4} e^{4y} y^{3/2} \big( 512y^3 + 960y^2 + 360y + 15 \big) \operatorname{erfc}(2\sqrt{y}) \Big],$$
$$\mathcal{C}^{(1)}_{USp(8)}(\tau, \bar{\tau}) = e^{2\pi i \tau} \frac{3}{2} \Big[ y^2 \big( 512y^3 + 1728y^2 + 1480y + 279 \big)$$
$$- \frac{\sqrt{\pi}}{4} e^{4y} y^{3/2} \big( 4096y^4 + 14336y^3 + 13440y^2 + 3360y + 105 \big) \operatorname{erfc}(2\sqrt{y}) \Big]. \tag{91}$$

We have verified that the above results, as well as the one-instanton contributions to $\mathcal{C}^{(1)}_{USp(2N)}(\tau, \bar{\tau})$ for other values of $N$ deduced from the Laplace-difference equation, again agree with the localisation computation in section 2.2.

Furthermore, from examples such as those in (91) we see that the one-instanton contributions to $\mathcal{C}_{USp(2N)}(\tau, \bar{\tau})$ for general values of $N$ in the weak coupling expansion behave as

$$\mathcal{C}^{(1)}_{USp(2N)}(\tau, \bar{\tau}) \sim e^{2\pi i \tau} \left( y^{1-N} + O(y^{-N}) \right). \tag{92}$$

This property is also evident from the localisation result, (37), which implies

$$\begin{aligned}
\mathcal{C}^{(1)}_{USp(2N)}(\tau, \bar{\tau}) &= \frac{1}{4} \Delta_\tau \left[ e^{2\pi i \tau} \left\langle \frac{1}{2} \prod_{j=1}^{N} \frac{a_j^2}{a_j^2 + 2} \right\rangle \right] \\
&\sim \Delta_\tau [ e^{2\pi i \tau} \left( y^{-N} + O(y^{-N-1}) \right) ] \sim e^{2\pi i \tau} \left( y^{1-N} + O(y^{-N}) \right).
\end{aligned} \tag{93}$$

The behaviour (92) implies that the one-instanton contribution to $\mathcal{C}_{USp(2N)}(\tau, \bar{\tau})$ is exponentially suppressed in the large-$N$ expansion. In fact, as we will see in the next section, for $USp(2N)$ all the contributions of odd instanton number are suppressed in the large-$N$ limit. This is in agreement with semi-classical instanton calculations based on the ADHM construction in [28].

The localisation computations of the $k$-instanton contributions with $k > 1$ to $\mathcal{C}_{SO(n)}$ and $\mathcal{C}_{USp(2N)}$ are not as explicitly understood as in the case of $\mathcal{C}_{SU(N)}$, due to complications in obtaining explicit expressions for the $k$-instanton Nekrasov partition functions. In section 2.2 we computed the two-instanton contributions to the Nekrasov partition function for the $SO(4)$ and $SO(5)$ theories, and the two- and three-instanton contributions to the $USp(2N)$ theories for $N \leq 5$, using the formulation given in [19, 20]. We have verified that all these multiple-instanton results, which originate from the localisation for the integrated correlators, agree with the exact formulae described in this section, and they provide further strong evidence to the validity of our conjecture (3).

# 5 The large-$N$ expansion

As in [1, 2] we will consider two distinct large-$N$ limits: one of these is a generalisation of the standard 't Hooft limit of the $SU(N)$ theory, in which $g_{YM}^2 N$ is fixed so that $g_{YM}^2 \sim 1/N$, and the contributions of Yang–Mills instantons are exponentially suppressed. The other large-$N$ limit is one with finite $g_{YM}^2$, in which instantons contribute and S-duality is manifest.

## 5.1 The 't Hooft limit

This is the limit in which the correlators have topological expansions reminiscent of 't Hooft's analysis of $SU(N)$ Yang–Mills theory in the large-$N$ limit [22]. However, the details of our analysis depend rather sensitively on whether $g_{YM}^2 N \ll 1$ or $g_{YM}^2 N \gg 1$. We will consider each in turn.

**The weakly coupled 't Hooft limit**

In this large-$N$ limit, the correlator $\mathcal{C}_{G_N}(\tau, \bar{\tau})$ is dominated by the perturbative contribution $\mathcal{C}^{pert}_{G_N}(\tau_2)$ (14), which has an expansion in powers of $a_{G_N}$ (defined in (16)), that is given by

$$\mathcal{C}_{G_N}(\tau, \bar{\tau}) \sim \mathcal{C}^{pert}_{G_N}(\tau_2) \sim c_{G_N} \sum_{g=0}^{\infty} (N_{G_N})^{-g} \mathcal{C}^{(g)}_{G_N}(a_{G_N}), \tag{94}$$

where $c_{G_N}$ is the central charge given in (18) and the parameters $N_{G_N}$ were defined in (22).

As emphasised in section 2.1, an interesting property of the integrated correlator is that its planar limit is identical for all the gauge groups. Therefore, one may simply use the known all-order planar result of $\mathcal{C}^{(0)}_{SU(N)}$ [2] and obtain

$$\mathcal{C}^{(0)}_{G_N}(a_{G_N}) = \sum_{m=1}^{\infty} \frac{(-4)^{m+1}\zeta(2m+1)\Gamma\left(m+\frac{3}{2}\right)^2}{\pi\Gamma(m)\Gamma(m+3)}(a_{G_N})^m. \tag{95}$$

This sum converges for $|a_{G_N}| < \frac{1}{4}$, and one can perform the convergent sum and obtain

$$\mathcal{C}^{(0)}_{G_N}(a_{G_N}) = a_{G_N} \int_0^{\infty} dw\, w^3 \frac{{}_1F_2\left(\frac{5}{2}; 2, 4 \,\middle|\, -4w^2 a_{G_N}\right)}{\sinh^2(w)}, \tag{96}$$

in agreement with [2,3]) and, the results given in [10] (after they are simplified). Sub-leading coefficients for each gauge group, $\mathcal{C}^{(g)}_{G_N}$ with $g \geq 1$, can be determined by using the Laplace-difference equations to any desired order (and agree with the sub-leading terms listed in [10] for each gauge group).

**The strongly coupled 't Hooft limit**

We now turn to the large-$N$ expansion of $\mathcal{C}_{G_N}(\tau, \bar{\tau})$ in the regime in which 't Hooft coupling is large. Once again this is a topological series analogous to (94). Whereas in the $SU(N)$ case the 't Hooft coupling is defined by $\lambda_{SU(N)} := g_{YM}^2 N$ the holographic connection with superstring amplitudes suggests that the strong-coupling 't Hooft parameter takes a somewhat different form in terms of $N$ in the case of $\mathcal{C}_{SO(2N)}, \mathcal{C}_{SO(2N+1)}$ and $\mathcal{C}_{USp(2N)}$. The holographic interpretation for general classical Lie groups [29] will be briefly reviewed in appendix D where it will be seen that the natural definition of the expansion coefficients for the various groups take the form

$$\lambda_{SU(N)} := g_{YM}^2 N, \qquad \lambda_{SO(n)} := g_{YM}^2\left(\frac{n}{2} - \frac{1}{4}\right), \qquad \lambda_{USp(n)} := g_{YM}^2\left(\frac{n}{2} + \frac{1}{4}\right), \tag{97}$$

which are in accord with [10] and are of the form $\lambda_{G_N} := g_{YM}^2 \tilde{N}_{G_N}$, where $\tilde{N}_{G_N}$ is the RR five-form flux in the appropriate orientifold background given by

$$\tilde{N}_{SU(N)} := N, \qquad \tilde{N}_{SO(n)} := \frac{n}{2} - \frac{1}{4}, \qquad \tilde{N}_{USp(n)} := \frac{n}{2} + \frac{1}{4}, \tag{98}$$

as discussed in appendix D. We see that, with the exception of the $SU(N)$ case, the $\lambda_{G_N}$ are different from $a_{G_N}$, which were the expansion parameters relevant in the weak coupling region and defined in (16).

With these definitions of the parameters we find that in the strong 't Hooft coupling region the large-$\tilde{N}_{G_N}$ asymptotic expansion of the integrated correlators takes the following form

$$\mathcal{C}_{G_N}(\lambda) \sim \sum_{g=0}^{\infty} (\tilde{N}_{G_N})^{2-2g} f^{(g)}_{G_N}(\lambda_{G_N}). \tag{99}$$

For each value of $g$ the asymptotic expansion of $f^{(g)}_{G_N}(\lambda_{G_N})$ in the large-$\lambda_{G_N}$ limit has the form of

$$f^{(g)}_{G_N}(\lambda_{G_N}) \sim \sum_{\ell} b^{(g)}_{\ell} \lambda_{G_N}^{-\ell/2}. \tag{100}$$

The $g = 0$ term, $f^{(0)}_{G_N}(\lambda_{G_N})$, can be obtained from (96) by expressing $a_{G_N}$ in terms of $\lambda_{G_N}$ and expanding for large $\lambda_{G_N}$. For $SU(N)$, we simply have $\lambda_{SU(N)} = 4\pi^2 a_{SU(N)} = g_{YM}^2 N$. For the

other classical gauge groups the the relations between $a_{G_N}$ and $\lambda_{G_N}$ are also simple in the large-$\tilde{N}_{G_N}$ limit, where to leading order we have

$$\lambda_{SO(n)} = 2\pi^2 a_{SO(n)} + O(n^{-1}), \qquad \lambda_{USp(n)} = 4\pi^2 a_{USp(n)} + O(n^{-1}). \qquad (101)$$

Using these relations and given that the planar contributions are identical for all gauge groups, the large-$\lambda_{G_N}$ expansions are determined by the $SU(N)$ results [2, 3]. We find

$$f_{USp(n)}^{(0)}(\lambda) = f_{SO(n)}^{(0)}(2\lambda) = \frac{1}{8} + \sum_{m=1}^{\infty} \frac{2^{1-2m}\Gamma\left(m-\frac{3}{2}\right)\Gamma\left(m+\frac{3}{2}\right)\Gamma(2m+1)\zeta(2m+1)}{\pi\,\Gamma(m)^2}\lambda^{-m-\frac{1}{2}}, \quad (102)$$

where the factor of 2 in the argument of $f_{SO(n)}^{(0)}(2\lambda)$ originates with (101). The sub-leading terms (the first few of which were determined in [10]) can also be determined in a systematic manner from the Laplace-difference equations. They have a structure that corresponds to terms that would arise in the low energy expansion of type IIB superstring amplitudes in an $AdS_5 \times S^5/\mathbb{Z}_2$ orientifold background.

In [2] it was shown that the large-$\lambda$ expansion of $f_{SU(N)}^{(g)}(\lambda)$ is an asymptotic series, which is not Borel summable. The analysis was carried out for $g=0$ and $g=1$ but in all likelihood it extends to all values of $g$. Applying ideas from resurgence similar to [2, 30–32], the large-$\lambda$ expansion of the correlator $\mathcal{C}_{SU(N)}(\tau, \bar\tau)$ therefore receives non-perturbative contributions, which behave as $e^{-\alpha\sqrt{\lambda}}$ for some constant $\alpha$. The same considerations appear in $\mathcal{C}_{SO(n)}$ and $\mathcal{C}_{USp(n)}$. In particular, the $g = 0$ terms in (102) take the same form as in $f_{SU(N)}^{(0)}(\lambda)$, and therefore have the same non-perturbative contributions. Similarly, the sub-leading powers of $\tilde{N}_{G_N}$ (terms with with $g > 0$ in (99)), have large-$\lambda$ expansions with very similar structures for all classical gauge groups. Once again, they are not Borel summable and are expected to have similar non-perturbative completions. It would be interesting to understand the path-integral semi-classical origin of these non-perturbative corrections, which have a behaviour suggestive of world-sheet instantons [2].

## 5.2 The fixed-$g_{YM}^2$ limit

In this limit the large-$N$ expansion of the integrated correlator is manifestly invariant under Montonen–Olive (or GNO) duality [2,7,8,10]. In order to determine an unlimited number of terms in this expansion we will combine the Laplace-difference equations with the results of the large-$N$ expansion of $\mathcal{C}_{SU(N)}(\tau, \bar\tau)$ determined in [2,7], which are summarised up to order $N^{-\frac{11}{2}}$ as follows

$$\mathcal{C}_{SU(N)}(\tau, \bar\tau) \sim \frac{N^2}{4} - \frac{3N^{\frac{1}{2}}}{2^4}E(\tfrac{3}{2}; \tau, \bar\tau) + \frac{45N^{-\frac{1}{2}}}{2^8}E(\tfrac{5}{2}; \tau, \bar\tau) \qquad (103)$$

$$+ N^{-\frac{3}{2}}\Big[\frac{4725}{2^{15}}E(\tfrac{7}{2}; \tau, \bar\tau) - \frac{39}{2^{13}}E(\tfrac{3}{2}; \tau, \bar\tau)\Big] + N^{-\frac{5}{2}}\Big[\frac{99225}{2^{18}}E(\tfrac{9}{2}; \tau, \bar\tau) - \frac{1125}{2^{16}}E(\tfrac{5}{2}; \tau, \bar\tau)\Big]$$

$$+ N^{-\frac{7}{2}}\Big[\frac{245581875}{2^{27}}E(\tfrac{11}{2}; \tau, \bar\tau) - \frac{2811375}{2^{25}}E(\tfrac{7}{2}; \tau, \bar\tau) + \frac{4599}{2^{22}}E(\tfrac{3}{2}; \tau, \bar\tau)\Big]$$

$$+ N^{-\frac{9}{2}}\Big[\frac{29499294825}{2^{31}}E(\tfrac{13}{2}; \tau, \bar\tau) - \frac{39590775}{2^{26}}E(\tfrac{9}{2}; \tau, \bar\tau) + \frac{1548855}{2^{27}}E(\tfrac{5}{2}; \tau, \bar\tau)\Big]$$

$$+ N^{-\frac{11}{2}}\Big[\frac{40266537436125}{2^{38}}E(\tfrac{15}{2}; \tau, \bar\tau) - \frac{397105891875}{2^{36}}E(\tfrac{11}{2}; \tau, \bar\tau) + \frac{2029052025}{2^{34}}E(\tfrac{7}{2}; \tau, \bar\tau)$$

$$- \frac{3611751}{2^{32}}E(\tfrac{3}{2}; \tau, \bar\tau)\Big] + O(N^{-\frac{13}{2}}).$$

As shown in [1,2], this result can be obtained directly from the large-$N$ expansion of (52). In these references, it was also shown that the Laplace-difference equation, (40) imposes strong

constraints on the form of (103). Thus, once the coefficients of the Eisenstein series with the highest values of $s$ at every power of $1/N$ (the 'highest-$s$' coefficients) are known, the Laplace-difference equation determines all the remaining expansion coefficients. But the highest-$s$ coefficients are completely determined by the planar-limit result obtained from (96), as shown in [2]. Therefore the large-$N$ expansion of the correlator is fully determined from (96) and the Laplace-difference equation.

Let us now consider the large-$N$ expansion of the integrated correlators of the $SO(n)$ theory using the Laplace-difference equation (41). We will solve the equation order by order in $1/\tilde{N}_{SO(n)}$, using the input of the large-$N$ expansion of $\mathcal{C}_{SU(N)}$, that was reviewed in the previous paragraph. We begin by making an ansatz for the the large-$\tilde{N}_{SO(n)}$ expansion of the $\mathcal{C}_{SO(n)}(\tau, \bar{\tau})$,

$$\mathcal{C}_{SO(n)}(\tau, \bar{\tau}) \sim (2\tilde{N}_{SO(n)})^2 \tilde{f}_2(\tau, \bar{\tau}) + (2\tilde{N}_{SO(n)})\tilde{f}_1(\tau, \bar{\tau}) + \tilde{f}_0(\tau, \bar{\tau}) + \sum_{\ell=0}^{\infty} (2\tilde{N}_{SO(n)})^{\frac{1}{2}-\ell} f_\ell(\tau, \bar{\tau}).$$

(104)

Here we choose to expand $\mathcal{C}_{SO(n)}(\tau, \bar{\tau})$ in powers of $2\tilde{N}_{SO(n)}$ in order to make the comparison with the expansion of $\mathcal{C}_{SU(N)}$ in (103) clearer. Substituting the ansatz (104) into the Laplace-difference equation (96) and expanding order by order in $1/n$ determines the equations satisfied by the coefficients of the powers of $\tilde{N}_{SO(n)}$. At order $n^2$, $n^1$ and $n^0$, the Laplace-difference equation leads to the conditions

$$\tilde{f}_2(\tau, \bar{\tau}) = \frac{1}{8}, \qquad \Delta_\tau \tilde{f}_1(\tau, \bar{\tau}) = \Delta_\tau \tilde{f}_0(\tau, \bar{\tau}) = 0.$$

(105)

Invariance under $SL(2, \mathbb{Z})$ implies $\tilde{f}_0(\tau, \bar{\tau})$ and $\tilde{f}_1(\tau, \bar{\tau})$ must be independent of $\tau$, and are therefore constant. Comparison with the perturbative expansion shows that these constants must each vanish. Indeed $\tilde{N}_{SO(n)}^2 \tilde{f}_2(\tau, \bar{\tau}) = \tilde{N}_{SO(n)}^2 /8$ precisely matches the supergravity expression.

The equations associated with half-integer powers in $n$ are more interesting. The Laplace-difference equation (96) implies that each coefficient function $f_\ell(\tau, \bar{\tau})$ must satisfy an inhomogeneous Laplace equation. The first such equation arises at order $n^{\frac{1}{2}}$ and takes the following form,

$$\left(\Delta_\tau + \frac{1}{4}\right) f_0(\tau, \bar{\tau}) = -\frac{3}{32} E\left(\tfrac{3}{2}; \tau, \bar{\tau}\right).$$

(106)

The above equation has the $SL(2, \mathbb{Z})$ invariant solution

$$f_0(\tau, \bar{\tau}) = -\frac{3}{32} E\left(\tfrac{3}{2}; \tau, \bar{\tau}\right) + \alpha\, E\left(\tfrac{1}{2}; \tau, \bar{\tau}\right).$$

(107)

The last term proportional to $E(\tfrac{1}{2}; \tau, \bar{\tau})$ is an arbitrary multiple of the modular invariant solution of the homogeneous equation

$$\left(\Delta_\tau + \frac{1}{4}\right) f_0(\tau, \bar{\tau}) = 0.$$

(108)

However, the coefficient $\alpha$ must vanish since the zero mode of $E\left(\tfrac{1}{2}; \tau, \bar{\tau}\right)$ is proportional to $\tau_2^{\frac{1}{2}} \log(\tau_2)$, which is inconsistent with the known perturbative result. Likewise, at order $n^{-\frac{1}{2}}$ we find the equation is given by

$$\left(\Delta_\tau - \frac{3}{4}\right) f_1(\tau, \bar{\tau}) = \frac{135}{512} E\left(\tfrac{5}{2}; \tau, \bar{\tau}\right),$$

(109)

which implies

$$f_1(\tau, \bar{\tau}) = \frac{45}{512} E\left(\tfrac{5}{2}; \tau, \bar{\tau}\right) + \beta\, E\left(\tfrac{3}{2}; \tau, \bar{\tau}\right),$$

(110)

where $E\left(\frac{3}{2}; \tau, \bar{\tau}\right)$ is the modular invariant solution of the homogeneous equation. However, either by comparing with the perturbative results [10] or with the one-instanton contributions presented in appendix B.4, we find $\beta = 0.$, so the coefficient of the inhomogeneous equation again vanishes. At order, $n^{-\frac{3}{2}}$ we find

$$\left(\Delta_\tau - \frac{15}{4}\right)f_2(\tau, \bar{\tau}) = \frac{23625}{2^{16}}E\left(\tfrac{7}{2}; \tau, \bar{\tau}\right) + \frac{333}{2^{14}}E\left(\tfrac{3}{2}; \tau, \bar{\tau}\right). \tag{111}$$

The $SL(2, \mathbb{Z})$-invariant solution to this equation is given by

$$f_2(\tau, \bar{\tau}) = \frac{4725}{2^{16}}E\left(\tfrac{7}{2}; \tau, \bar{\tau}\right) - \frac{111}{2^{14}}E\left(\tfrac{3}{2}; \tau, \bar{\tau}\right) + \gamma E\left(\tfrac{5}{2}; \tau, \bar{\tau}\right), \tag{112}$$

where $\gamma E\left(\frac{5}{2}; \tau, \bar{\tau}\right)$ is the modular invariant homogeneous solution, which again has to vanish in order to be consistent with the perturbative result or the one-instanton contribution.

One may proceed in a similar way to obtain the expressions for $f_\ell(\tau, \bar{\tau})$ for general $\ell$. For each value of $\ell$, the function $f_\ell(\tau, \bar{\tau})$ gets a contribution proportional to $E\left(\ell + \frac{1}{2}; \tau, \bar{\tau}\right)$ from the modular invariant solution of a homogeneous Laplace equation. Such a contribution must have vanishing coefficient since it is inconsistent with the structure (99). To see this we may substitute the relation $\tau_2 = 2\pi(n - \frac{1}{2})/\lambda_{SO(n)}$ into the zero mode of $E\left(\ell + \frac{1}{2}; \tau, \bar{\tau}\right)$ (the sum of the $\tau_2^{\ell+\frac{1}{2}}$ and $\tau_2^{-\ell+\frac{1}{2}}$ terms) to convert to the variables $\tilde{N}_{SO(n)}$ and $\lambda_{SO(n)}$. It is easy to see that such a contribution behaves as $\tilde{N}_{SO(n)}^{2-g}$ (instead of $\tilde{N}_{SO(n)}^{2-2g}$), which is inconsistent with the general structure given in (99). In particular, these solutions to the homogeneous equations would lead to perturbative terms proportional to $\tilde{N}_{SO(n)}$, which are not present in the perturbative computation [10].

We therefore conclude that all of the solutions to the homogeneous equations must have vanishing coefficients. This is similar to the systematics of the solution of the Laplace-difference equation of the $SU(N)$ correlator, as analysed in [2] where, at order $N^{\frac{1}{2}-\ell}$ the coefficient multiplying $E\left(\ell + \frac{3}{2}; \tau, \bar{\tau}\right)$ was not determined by the Laplace-difference equation.

Once the solutions to the homogeneous equations have been set to zero, the Laplace-difference equations determine the coefficients in the large-$\tilde{N}_{SO(n)}$ expansion uniquely. In this manner we find

$$
\begin{aligned}
2\mathcal{C}_{SO(n)}(\tau, \bar{\tau}) \sim{} & \frac{(2\tilde{N}_{SO(n)})^2}{4} - \frac{3(2\tilde{N}_{SO(n)})^{\frac{1}{2}}}{2^4}E(\tfrac{3}{2}; \tau, \bar{\tau}) + \frac{45(2\tilde{N}_{SO(n)})^{-\frac{1}{2}}}{2^8}E(\tfrac{5}{2}; \tau, \bar{\tau}) \\
& + (2\tilde{N}_{SO(n)})^{-\frac{3}{2}}\left[\frac{4725}{2^{15}}E(\tfrac{7}{2}; \tau, \bar{\tau}) - \frac{111}{2^{13}}E(\tfrac{3}{2}; \tau, \bar{\tau})\right] \\
& + (2\tilde{N}_{SO(n)})^{-\frac{5}{2}}\left[\frac{99225}{2^{18}}E(\tfrac{9}{2}; \tau, \bar{\tau}) - \frac{3825}{2^{16}}E(\tfrac{5}{2}; \tau, \bar{\tau})\right] \\
& + (2\tilde{N}_{SO(n)})^{-\frac{7}{2}}\left[\frac{245581875}{2^{27}}E(\tfrac{11}{2}; \tau, \bar{\tau}) - \frac{10749375}{2^{25}}E(\tfrac{7}{2}; \tau, \bar{\tau}) + \frac{40239}{2^{22}}E(\tfrac{3}{2}; \tau, \bar{\tau})\right] \\
& + (2\tilde{N}_{SO(n)})^{-\frac{9}{2}}\left[\frac{29499294825}{2^{31}}E(\tfrac{13}{2}; \tau, \bar{\tau}) - \frac{164614275}{2^{26}}E(\tfrac{9}{2}; \tau, \bar{\tau}) + \frac{18332055}{2^{27}}E(\tfrac{5}{2}; \tau, \bar{\tau})\right] \\
& + (2\tilde{N}_{SO(n)})^{-\frac{11}{2}}\left[\frac{40266537436125}{2^{38}}E(\tfrac{15}{2}; \tau, \bar{\tau})\right. \\
& \left. - \frac{1758611806875}{2^{36}}E(\tfrac{11}{2}; \tau, \bar{\tau}) + \frac{28855523025}{2^{34}}E(\tfrac{7}{2}; \tau, \bar{\tau}) \right. \\
& \left. - \frac{103062039}{2^{32}}E(\tfrac{3}{2}; \tau, \bar{\tau})\right] + O(\tilde{N}_{SO(n)}^{-\frac{13}{2}}). \tag{113}
\end{aligned}
$$

This expression applies to $\mathcal{C}_{SO(n)}$ for both $n = 2N$ and $n = 2N + 1$. As described earlier, we have presented the expansion as a series in $(2\tilde{N}_{SO(n)})^{-1}$ in order to emphasise similarities in

the coefficients with those of the expansion in the $SU(N)$ case, (103). Indeed, the highest-$s$ terms in the large-flux number expansion are identical for $\mathcal{C}_{SO(n)}(\tau, \bar{\tau})$ and $\mathcal{C}_{SU(N)}(\tau, \bar{\tau})$, apart from an overall factor of two. As we saw earlier, the coefficients of the Eisenstein series with highest index $s$ are determined by the planar limit. We also know that the planar contributions to the integrated correlators are identical to all gauge groups. These statements imply that the highest-$s$ terms are the same for all gauge groups.[14] If one re-expands (113) in powers of $c_{SO(n)}^{-1}$ instead of $\tilde{N}_{SO(n)}^{-1}$ the expansion agrees with the expressions in [10], which were computed up to $O(c_{SO(n)}^{-7/4})$. However, using the Laplace-difference equation makes it easy to obtain the expansion to any desired order.

We have also solved the Laplace-difference equation (42) for the coefficients in the large-$\tilde{N}_{USp(n)}$ expansion of $\mathcal{C}_{USp(n)}(\tau, \bar{\tau})$. At each order in $1/\tilde{N}_{USp(n)}$ the equation for $\mathcal{C}_{USp(n)}$ is identical to that of $\mathcal{C}_{SO(n)}$, except that the terms involving the $SU(N)$ correlators depend on the rescaled coupling, $(\tau, \bar{\tau}) \to (2\tau, 2\bar{\tau})$. Therefore, we find the result is identical to that of the $SO(n)$ theory, but with $(\tau, \bar{\tau}) \to (2\tau, 2\bar{\tau})$, and with $\tilde{N}_{SO(n)} \to \tilde{N}_{USp(n)}$, so that, in the large-$\tilde{N}_{G_N}$ expansion,

$$\mathcal{C}_{USp(n)}(\tau, \bar{\tau}) \sim \mathcal{C}_{SO(n)}(2\tau, 2\bar{\tau})\Big|_{\tilde{N}_{SO(n)} \to \tilde{N}_{USp(n)}}. \tag{114}$$

The rescaling $(\tau, \bar{\tau}) \to (2\tau, 2\bar{\tau})$ in this expression also implies that odd instanton number terms do not contribute to the integrated correlator of $USp(2N)$ in the large-$N$ expansion. In particular, the one-instanton contribution is suppressed, as we showed in (93) from the explicit one-instanton computation based on localisation. One can also see the suppression of the odd-number instantons from the general expression of the integrated correlator given in (72),

$$\mathcal{C}_{USp(2N)}(\tau, \bar{\tau}) = \sum_{(m,n)\in\mathbb{Z}^2} \int_0^\infty dt \left( B_{USp(2N)}^1(t) e^{-t\pi \frac{|m+n\tau|^2}{\tau_2}} + B_{USp(2N)}^2(t) e^{-t\pi \frac{|m+2n\tau|^2}{2\tau_2}} \right), \tag{115}$$

and recall using (79) that

$$B_{USp(2N)}^1(t) = B_{SO(2N+1)}^2(t) = N(2N+1)\frac{t\left(3t^2 - (8N+2)t + 3\right)}{2(t-1)^2(t+1)^3}\left(\frac{t-1}{t+1}\right)^{2N}. \tag{116}$$

Following a similar analysis to that given in [2], one can see that in the large-$N$ limit, the contribution to (116) from $B_{USp(2N)}^1(t)$ is a coupling-independent constant, with corrections that are exponentially suppressed. This can be seen as follows. The $k$-instanton contribution arising from $B_{USp(2N)}^1(t)$ can be expressed via a Poisson summation in the form

$$e^{2\pi i k \tau_1} \sqrt{\tau_2} \sum_{\hat{m}\neq 0, n\neq 0} \int_0^\infty \frac{dt}{\sqrt{t}} B_{USp(2N)}^1(t) e^{-\hat{m}^2 \pi \tau_2/t - n^2 \pi \tau_2 t}, \tag{117}$$

with $\hat{m}n = k$. This is suppressed because the last factor in (116) satisfies $\left(\frac{t-1}{t+1}\right)^{2N} < 1$ in the integration region $0 < t < \infty$, apart from the boundaries at $t = 0$ and $t = \infty$ (which are however are also suppressed due to the exponential terms $e^{-\hat{m}^2 \pi \tau_2/t}$ and $e^{-n^2 \pi \tau_2 t}$ in (117), respectively). Similarly, one can show that the perturbative (i.e. zero-instanton) contribution is also exponentially suppressed in the large-$N$ limit apart from a coupling independent constant.

Therefore, only the second term in (115) survives in the large-$N$ expansion (apart from the coupling-independent constant mentioned above). This means that $\mathcal{C}_{USp(2N)}(\tau, \bar{\tau})$ only gets contributions from terms with an even number of instantons, which is in accord with the

---

[14]The overall factor of 2 is due to the fact that $c_{SU(N)} \sim \frac{1}{4}\tilde{N}_{SU(N)}$, while $c_{SO(n)} \sim \frac{1}{8}\tilde{N}_{SO(n)}$ in the large-$\tilde{N}_{G_N}$ limit.

calculation in [28] of the leading $k$-instanton contribution to the large-$N$ limit based on the ADHM construction. Here we see this is true to all orders in large-$N$ expansion. Using (72) and (73), and the analysis discussed above, we find that

$$\mathcal{C}_{USp(2N)}(\tau, \bar{\tau}) \sim \mathcal{C}_{SO(2N+1)}(2\tau, 2\bar{\tau}). \tag{118}$$

This is in agreement with our earlier findings (114) since $\tilde{N}_{SO(2N+1)} = \tilde{N}_{USp(2N)}$.

The structure of (103), (113) and (114) extend the $SU(N)$ results in [7] and the $SO(n)$ and $USp(n)$ results in [10]. A notable feature of the structure of these large-$N$ expressions is the fact that the Eisenstein series that arise at each order in $1/\tilde{N}_{G_N}$ have half-integer index, whereas those that arise at finite $N$ in (8) have integer index. The low order terms in the large-$N$ expressions have a close connection to corresponding BPS terms in the low energy expansion of the holographically dual type IIB superstring amplitudes, as described in the earlier references.

# 6 Discussion

In this paper we have proposed a lattice sum representation of the integrated correlator, $\mathcal{C}_{G_N}(\tau, \bar{\tau})$, of four superconformal primary operators in the stress tensor multiplet in $\mathcal{N} = 4$ SYM that are defined by (1) with any classical gauge group. This generalises the expression proposed for $SU(N)$ gauge groups in [1,2]. Such integrated correlators, which are determined by supersymmetric localisation, are highly constrained by maximal supersymmetry and satisfy a fascinating interplay of properties that reflects the constraints imposed by GNO duality.

There are several obvious directions in which these ideas could be extended. A challenging objective would be to extend the discussion in this paper to $\mathcal{N} = 4$ SYM with exceptional gauge groups. These are theories that are self-dual under the action of GNO S-duality. With gauge groups $E_6$, $E_7$ and $E_8$, which are simply-laced, the duality group is $SL(2, \mathbb{Z})$. However, the duality groups in the non simply-laced cases, $G_2$ and $F_4$, are Hecke groups, which have novel features that will not be reproduced in terms of non-holomorphic Eisenstein series. Since supersymmetric localisation is ill-understood for exceptional groups an alternative procedure is needed, perhaps making use of the modular anomaly equation, as suggested in [33]. Another challenge is to construct expressions for integrated $n$-point correlators with $n > 4$. Although the general problem is daunting, following the methods of this paper and the previous results of [34,35], it should be possible to obtain exact expressions for integrated maximal $U(1)_Y$-violating $n$-point correlators with $n > 4$ for all classical gauge groups, which transform covariantly under GNO S-duality (where $U(1)_Y$ is the bonus symmetry [36]).

Another interesting direction is to formulate lattice representations for other integrated correlators. In particular, the correlator given by $\partial_m^4 \log Z(m, \tau, \bar{\tau})|_{m=0}$ was analysed in the large-$N$ limit for $SU(N)$ gauge groups, in [6,8]. In that case the coefficients of integer powers of $1/N$ are generalised Eisenstein series that satisfy inhomogeneous Laplace eigenvalue equations with sources terms that are quadratic in non-holomorphic Eisenstein series. It would be of interest to discover the structure of such correlators at finite values of $N$, perhaps using the recent results of [37,38], and for more general gauge groups.

More generally, it should be of interest to consider integrated correlators in a wider context. While integration over the operator insertion points obviously averages over the detailed form of any correlator, it remains uncertain as to how much information may be retrieved by considering the set of all possible integrated correlators. Clearly, supersymmetry has played a crucial rôle in constraining the integrated correlators we have considered, so it would be interesting to understand how deformations that break supersymmetry affect their structure.

Finally, it would be of interest to understand the extent to which properties of the integrated correlators can be used as probes of the fundamental structure of string theory.

## Acknowledgements

We would like to thank Shai Chester, Stefano Cremonesi, Scott Collier, Nick Dorey, Francesco Fucito, Iñaki García-Etxebarria, Amihay Hanany, Francisco Morales and Eric Perlmutter for useful conversations. CW is supported by a Royal Society University Research Fellowship No. UF160350.

# A   Goddard–Nuyts–Olive duality

The 1977 paper by Goddard, Nuyts and Olive [14] showed that while electric charges in gauge theories take their values in the weight lattice of the gauge group $G$, the magnetic charges take their values in the lattice of a dual group $^{L}G$. Table 1 lists the dual groups corresponding to each of the classical Lie groups. Montonen and Olive [11], conjectured that there is a duality that identifies a gauge theory with gauge group $G$ and coupling $g_{YM}$ with a theory with gauge group $^{L}G$ and coupling $^{L}g_{YM} = 4\pi/g_{YM}$. The rôles of electric and magnetic charges are interchanged by this duality. It was later understood [12] that such a duality requires supersymmetry, and in 1979 it was argued [13] that this duality could be realised in $\mathcal{N} = 4$ SYM in which the $\mathbb{Z}_2$ inversion of the coupling is naturally extended to $SL(2,\mathbb{Z})$ acting on the complex coupling $\tau = \frac{\theta}{2\pi} + i\frac{4\pi}{g_{YM}^2}$ and the spectrum contains infinite towers of dyonic states carrying both electric and magnetic charge.

Table 1:   Langlands/GNO relation between classiical Lie groups and their dual groups.

| $G_N$ | $^{L}G_N$ |
|---|---|
| $U(N)$ | $U(N)$ |
| $SU(N)$ | $PSU(N) = SU(N)/\mathbb{Z}_N$ |
| $Spin(2N)$ | $SO(2N)/\mathbb{Z}_2$ |
| $Sp(N) = USp(2N)$ | $SO(2N+1)$ |
| $Spin(2N+1)$ | $Sp(N)/\mathbb{Z}_2 = USp(2N)/\mathbb{Z}_2$ |
| $G_2$ | $G_2$ |
| $F_4$ | $F_4$ |
| $E_{r=6,7,8}$ | $E_r/\mathbb{Z}_{9-r}$ |

This story has close connections to the Langlands programme [17] and the GNO dual group is identified with the Langlands dual group (hence the superscript on $^{L}G$). The extensive connections between the geometric Langlands programme and the dualities of $\mathcal{N} = 4$ SYM is explored in [17] and subsequent papers.

The integrated correlators that are the subject of this paper are not sensitive to the discrete stability groups of $^{L}G$ listed in the right-hand column of table 1. They also do not distinguish

between $Spin(N)$ and $SO(N)$. This means that the discussions in this paper are at the level of the Lie algebra, $\mathfrak{g}_N$, as shown by the labels in table 2 .

The S-duality transformation maps a theory with gauge group $G$ into one with gauge group $^LG$. The Montonen–Olive inversion of the coupling constant, $\tau_2 \to \tau_2^{-1}$ generalises to the $\hat{S}$ and $T$ transformations, which are defined by

$$T : (G_N, \tau) \to (G_N, \tau + 1),$$
$$\hat{S} : (G_N, \tau) \to (^LG_N, -\frac{1}{r\tau}), \tag{119}$$

where $r$ is the square of the ratio of the long and short roots of the Lie algebra of $G_N$. In the simply laced cases, i.e. $SU(N)$ and $SO(2N)$, $r = 1$ and $\hat{S} \equiv S : \tau \to -1/\tau$ reduces to the Montonen–Olive transformation when $\tau_1 = 0$. In these cases $S$ and $T$ are the generators of the discrete self-duality group $SL(2, \mathbb{Z})$, under which

$$\tau \underset{SL(2,\mathbb{Z})}{\to} \frac{a\tau + b}{c\tau + d}, \tag{120}$$

where $a, b, c, d \in \mathbb{Z}$ with $ad - bc = 1$.

Table 2: Duality relations of relevance to this paper.

| $\mathfrak{g}_N$ | $^L\mathfrak{g}_N$ |
|---|---|
| $su(N)$ | $su(N)$ |
| $so(2N)$ | $so(2N)$ |
| $usp(2N)$ | $so(2N+1)$ |
| $so(2N+1)$ | $usp(2N)$ |

In the non simply-laced cases of interest to us $r = 2$ and the $\hat{S}$ transformation $\tau \to -1/(2\tau)$ maps theories with gauge groups $SO(2N+1)$ and $USp(2N)$ into each other.[15] In these cases $\hat{S}$ generates an $SL(2, \mathbb{R})$ transformation that is not in $SL(2, \mathbb{Z})$. It is easy to see that the operators $\hat{S}T\hat{S}$ and $T$ generate a $\Gamma_0(r)$ subgroup of $SL(2, \mathbb{Z})$ (which is a subgroup in which $c = 0 \bmod r$). In other words $\Gamma_0(2)$ is a self-duality group that maps $\mathcal{C}_{G_N}$ into $\mathcal{C}_{G_N}$ and $\mathcal{C}_{{}^LG_N}$ into $\mathcal{C}_{{}^LG_N}$.

There are a number of distinctive features involved in S-duality for gauge theories with exceptional groups [17,39]. In the simply-laced cases ($E_6, E_7$ and $E_8$). S-duality is a symmetry associated with the action of $SL(2, \mathbb{Z})$. In the non simply-laced cases ($F_4$ and $G_2$, which have $r = 2$ and $r = 3$, respectively) S-duality is again a symmetry, but the presence of both long and short roots implies that the duality group is a Hecke group rather than a subgroup of $SL(2, \mathbb{Z})$, which is generated by $\hat{S}$ and $T$.

# B  Integrated correlators from localisation

In this appendix, we will review the computation of the integrated correlators (1) using supersymmetric localisation. We begin with a brief review of the application of localisation to the calculation of integrated correlators.

---

[15]This ratio is $r = 1$ for the exceptional groups $E_6, E_7$ and $E_8$, while $r = 2$ for $F_4$, and $r = 3$ for $G_2$. All the exceptional groups are self-dual (possibly modulo some discrete quotient), i.e. $^LG = G$. when $G = G_2, F_4, E_6, E_7, E_8$.

## B.1 Review of integrated correlators

The starting point is the partition function of $\mathcal{N} = 2^*$ SYM on $S^4$, which was determined by Pestun using supersymmetric localisation in [4], where it was shown to have the form[16]

$$
\begin{aligned}
Z_{G_N}(m, \tau, \bar{\tau}) &= \frac{1}{\mathcal{N}_{G_N}} \int d^r a \, v_{G_N}(a) e^{-\frac{8\pi^2}{g_{YM}^2} \langle a, a \rangle} \hat{Z}_{G_N}^{pert}(m, a) |\hat{Z}_{G_N}^{inst}(m, \tau, a)|^2 \\
&= \langle Z_{G_N}^{pert}(m, a) |\hat{Z}_{G_N}^{inst}(m, \tau, a)|^2 \rangle_{G_N},
\end{aligned}
\tag{121}
$$

where the integration variable $a$ runs over the $r$-dimensional Cartan subalgebra of $G_N$, $v_{G_N}(a)$ is the Vandermonde determinant associated with the group $G_N$, and the Killing form $\langle a, a \rangle$ is equal to $\text{tr}_s(a\,a)/(2T_s)$, where $T_s$ is the Dynkin index and $s$ denotes the representation. The normalisation factor $\mathcal{N}_{G_N}$ is given by

$$
\mathcal{N}_{G_N} = \int d^r a \, v_{G_N}(a) e^{-\frac{8\pi^2}{g_{YM}^2} \langle a, a \rangle}.
\tag{122}
$$

We see from (121) that the expectation value of a general function $F(a_i)$ is defined by

$$
\langle F(a_i) \rangle_{G_N} = \frac{1}{\mathcal{N}_{G_N}} \int d^r a \, v_{G_N}(a) e^{-\frac{8\pi^2}{g_{YM}^2} \langle a, a \rangle} F(a_i),
\tag{123}
$$

so that with the given definition for $\mathcal{N}_{G_N}$ above, we have $\langle 1 \rangle_{G_N} = 1$.

The perturbative contribution to the partition function is one-loop exact and is given by the classical factor proportional to $\exp(-8\pi^2 \langle a, a \rangle / g_{YM}^2)$ multiplying the one-loop term,

$$
\hat{Z}_{G_N}^{pert}(m, a) = \frac{1}{H(m)^r} \prod_{\alpha \in \Delta} \frac{H(\alpha \cdot a)}{\left[ H(\alpha \cdot a + m) H(\alpha \cdot a - m) \right]^{\frac{1}{2}}}.
\tag{124}
$$

Here $r$ denotes the rank of $G_N$, while the product runs over the set of roots. The function $H(z)$ is given by $H(z) = e^{-(1+\gamma)z^2} G(1+iz) G(1-iz)$, where $G(z)$ is Barnes G-function (and $\gamma$ is the Euler constant). The factor of $|\hat{Z}_{G_N}^{inst}|^2 = \hat{Z}_{G_N}^{inst} \bar{\hat{Z}}_{G_N}^{inst}$ in (121) is the contribution from the Nekrasov partition function and describes the contributions from instantons and anti-instantons localised at the north and south poles of $S^4$.

The integrated correlation functions of interest for the present paper were defined in [3] (for $G_N = SU(N)$) where they were obtained by acting on $\log Z_{G_N}$ with various derivatives with respect to the hypermultiplet mass, $m$, and the complex coupling, $\tau$, followed by the limit $m \to 0$, as displayed in (1). In the same reference [3], it was shown that this quantity is equal to the correlator of four superconformal primary operators of the stress tensor supermultiplet integrated over their positions with a specific measure that maintains supersymmetry.

The result may be separated into perturbative and instanton contributions since

$$
\partial_m^2 \log Z_{G_N}\big|_{m=0} = \partial_m^2 \log Z_{G_N}^{pert}\big|_{m=0} + \partial_m^2 \log Z_{G_N}^{inst}\big|_{m=0},
\tag{125}
$$

where each contribution can be expressed as an expectation value in a gaussian matrix model,

$$
\partial_m^2 \log Z_{G_N}^{pert}\big|_{m=0} = \langle \partial_m^2 \hat{Z}_{G_N}^{pert}\big|_{m=0} \rangle_{G_N}, \quad \partial_m^2 \log Z_{G_N}^{inst}\big|_{m=0} = \langle \partial_m^2 \hat{Z}_{G_N}^{inst}\big|_{m=0} \rangle_{G_N}.
\tag{126}
$$

The gaussian model expectation value, $\langle \dots \rangle_{G_N}$, is defined by (123) and its explicit form for each gauge group is given in appendix B.2, where the expressions for the perturbative parts of $\mathcal{C}_{G_N}$ determined in [10] are reviewed. A review of the general structure of the instanton contributions that were discussed in [19, 20], is given in appendix B.3.

---

[16]The subscript on $Z_{G_N}$ indicates that the gauge group is $G_N$.

## B.2 Perturbative contributions

The discussion in [10] focussed on the perturbative sector, where $\hat{Z}_{G_N}^{inst} = 1$, and where the partition function has no dependence on $\tau_1 = \theta/(2\pi)$. In this subsection we will review the explicit form of this measure, as well as the expressions for $\hat{Z}_{G_N}^{pert}$, given in [10] for each classical gauge group.

- $SU(N)$

$$\hat{Z}_{SU(N)}^{pert}(m, a_i) = \frac{1}{H(m)^{N-1}} \prod_{i<j} \frac{H^2(a_{ij})}{H(a_{ij} + m)H(a_{ij} - m)}, \tag{127}$$

where $a_{ij} = a_i - a_j$. The expectation value of any function $F(a_i)$ in the $SU(N)$ case is obtained from (123) and has the form

$$\langle F(a_i) \rangle_{SU(N)} = \frac{1}{\mathcal{N}_{SU(N)}} \int d^N a \, \delta\left(\sum_i a_i\right) \prod_{i<j} a_{ij}^2 \, e^{-\frac{8\pi^2}{g_{YM}^2} \sum_i a_i^2} F(a_i). \tag{128}$$

- $SO(2N)$

$$\hat{Z}_{SO(2N)}^{pert}(m, a_i) = \frac{1}{H(m)^N} \prod_{i<j} \frac{H^2(a_{ij}) H^2(a_{ij}^+)}{H(a_{ij} + m)H(a_{ij} - m)H(a_{ij}^+ + m)H(a_{ij}^+ - m)}, \tag{129}$$

where $a_{ij}^+ = a_i + a_j$. The expectation value of $F(a_i)$ in the $SO(2N)$ case is given by the integral

$$\langle F(a_i) \rangle_{SO(2N)} = \frac{1}{\mathcal{N}_{SO(2N)}} \int d^N a \prod_{i<j} a_{ij}^2 (a_{ij}^+)^2 \, e^{-\frac{8\pi^2}{g_{YM}^2} \sum_i a_i^2} F(a_i). \tag{130}$$

- $SO(2N+1)$

$$\begin{aligned}
\hat{Z}_{SO(2N+1)}^{pert}(m, a_i) = & \frac{1}{H(m)^N} \prod_i \frac{H^2(a_i)}{H(a_i + m)H(a_i - m)} \\
& \prod_{i<j} \frac{H^2(a_{ij}) H^2(a_{ij}^+)}{H(a_{ij} + m)H(a_{ij} - m)H(a_{ij}^+ + m)H(a_{ij}^+ - m)}.
\end{aligned} \tag{131}$$

The expectation value of $F(a_i)$ in the $SO(2N+1)$ case is given by the integral

$$\langle F(a_i) \rangle_{SO(2N+1)} = \frac{1}{\mathcal{N}_{SO(2N+1)}} \int d^N a \prod_i a_i^2 \prod_{i<j} a_{ij}^2 (a_{ij}^+)^2 \, e^{-\frac{8\pi^2}{(\delta_{N,1}+1)g_{YM}^2} \sum_i a_i^2} F(a_i). \tag{132}$$

- $USp(2N)$

$$\begin{aligned}
\hat{Z}_{USp(2N)}^{pert}(m, a_i) = & \frac{1}{H(m)^N} \prod_i \frac{H^2(2a_i)}{H(2a_i + m)H(2a_i - m)} \\
& \prod_{i<j} \frac{H^2(a_{ij}) H^2(a_{ij}^+)}{H(a_{ij} + m)H(a_{ij} - m)H(a_{ij}^+ + m)H(a_{ij}^+ - m)}.
\end{aligned} \tag{133}$$

The expectation value of $F(a_i)$ in the $USp(2N)$ case is given by the integral

$$\langle F(a_i) \rangle_{USp(2N)} = \frac{1}{\mathcal{N}_{USp(2N)}} \int d^N a \prod_i a_i^2 \prod_{i<j} a_{ij}^2 (a_{ij}^+)^2 \, e^{-\frac{16\pi^2}{g_{YM}^2} \sum_i a_i^2} F(a_i). \tag{134}$$

The perturbative contributions to $\mathcal{C}_{G_N}(\tau, \bar{\tau})$ form an essential ingredient in our discussion. They are given by substituting the above expressions into (1), which leads to the following expressions that are given in equation (3.8) of [10],[17]

$$\mathcal{C}_{SU(N)}^{pert}(y) = -\int_0^\infty d\omega \frac{\omega}{2\sinh^2\omega} y^2 \partial_y^2 \sum_{i,j=1}^N e^{-\frac{\omega^2}{y}}\left[ L_{i-1}\left(\frac{\omega^2}{y}\right)L_{j-1}\left(\frac{\omega^2}{y}\right) \right.$$
$$\left. -(-1)^{i-j}L_{i-1}^{j-i}\left(\frac{\omega^2}{y}\right)L_{j-1}^{i-j}\left(\frac{\omega^2}{y}\right) \right], \tag{135}$$

$$\mathcal{C}_{SO(2N)}^{pert}(y) = -\int_0^\infty d\omega \frac{\omega}{\sinh^2\omega} y^2 \partial_y^2 \sum_{i,j=1}^N e^{-\frac{\omega^2}{y}}\left[ L_{2(i-1)}\left(\frac{\omega^2}{y}\right)L_{2(j-1)}\left(\frac{\omega^2}{y}\right) \right.$$
$$\left. -L_{2(i-1)}^{2(j-i)}\left(\frac{\omega^2}{y}\right)L_{2(j-1)}^{2(i-j)}\left(\frac{\omega^2}{y}\right) \right], \tag{136}$$

$$\mathcal{C}_{SO(2N+1)}^{pert}(y) = -\int_0^\infty d\omega \frac{\omega}{\sinh^2\omega} y^2 \partial_y^2 \left\{ e^{-\frac{\omega^2}{y}} \sum_{i,j=1}^N \left[ L_{2i-1}\left(\frac{\omega^2}{y}\right)L_{2j-1}\left(\frac{\omega^2}{y}\right) \right. \right.$$
$$\left. \left. -L_{2i-1}^{2(j-i)}\left(\frac{\omega^2}{y}\right)L_{2j-1}^{2(i-j)}\left(\frac{\omega^2}{y}\right) \right] + e^{-\frac{\omega^2}{2y}} \sum_{i=1}^N L_{2i-1}\left(\frac{\omega^2}{y}\right) \right\}, \tag{137}$$

$$\mathcal{C}_{USp(2N)}^{pert}(y) = -\int_0^\infty d\omega \frac{\omega}{\sinh^2\omega} y^2 \partial_y^2 \left\{ e^{-\frac{\omega^2}{2y}} \sum_{i,j=1}^N \left[ L_{2i-1}\left(\frac{\omega^2}{2y}\right)L_{2j-1}\left(\frac{\omega^2}{2y}\right) \right. \right.$$
$$\left. \left. -L_{2i-1}^{2(j-i)}\left(\frac{\omega^2}{2y}\right)L_{2j-1}^{2(i-j)}\left(\frac{\omega^2}{2y}\right) \right] + e^{-\frac{\omega^2}{y}} \sum_{i=1}^N L_{2i-1}\left(\frac{2\omega^2}{y}\right) \right\}, \tag{138}$$

where $y = \pi\tau_2 = 4\pi^2/g_{YM}^2$, and $L_n^\alpha(x)$ are generalized Laguerre polynomials.[18] When $\alpha = 0$ one recovers the standard Laguerre polynomials, $L_n(x) := L_n^0(x)$. For any fixed value of $N$ the above expressions can be expanded in powers of $g_{YM}^2$ to generate the perturbation expansions shown in (17).

We note the following:

- The $SO(2N+1)$ result only holds for $N > 1$ and the $SO(3)$ case is special since the Dynkin index of $SO(n)$ is discontinuous as $n = 3$ is changed to $n > 3$. For $SO(3)$ we must rescale the coupling constant inside the square brackets in (137) by $g_{YM} \to \sqrt{2}g_{YM}$. With this rescaling the correlator $\mathcal{C}_{SO(3)}$ is identical to $\mathcal{C}_{SU(2)}$.

- These formulae satisfy the isomorphisms $SU(2) \cong SO(3) \cong USp(2)$, $SU(4) \cong SO(6)$, $SO(4) \cong SU(2) \times SU(2)$, and $SO(5) \cong USp(4)$.

## B.3  Instanton contributions

Much of this section is a review of [19, 20]. The Nekrasov partition functions that describe the instanton contributions are expressed as infinite Fourier sums,

$$\hat{Z}_{G_N}^{inst}(m, \tau, a_i) = \sum_{k=0}^\infty e^{2\pi i k \tau} \hat{Z}_{G_N}^{(k)}(m, a_i), \tag{139}$$

---

[17]The $SU(N)$ case was determined in [5]. Furthermore, the expressions in [10] have been multiplied by a factor of 4 to accord with our conventions.

[18]Laguerre polynomials have previously appeared in the perturbative sector of Wilson loop calculations in these theories [40].

where $k$ is the number of instantons, and $\hat{Z}_{G_N}^{(k)\,inst}(m, a_i)$ can be conveniently expressed as a contour integral,

$$\hat{Z}_{G_N}^{(k)}(m, a_i) = \oint \prod_{I=1}^{\ell} \frac{d\phi_I}{2\pi} \widetilde{Z}_{G_N}^{(k)\,gauge}(m, a_i, \phi_I) \widetilde{Z}_{G_N}^{(k)\,matter}(m, a_i, \phi_I), \qquad (140)$$

where $\ell = k$ for $SU(N), SO(2N)$ and $SO(2N+1)$, while for $USp(2N)$, $\ell = K = \left\lfloor \frac{k}{2} \right\rfloor$. The expressions for $\widetilde{Z}_{G_N}^{(k)\,gauge}$ and $\widetilde{Z}_{G_N}^{(k)\,matter}$ for each group will be summarised below. In the $SU(N)$ case the contour integral was performed explicitly and a general expression for $\partial_m^2 \hat{Z}_{SU(N)}^{(k)}|_{m=0}$ was obtained in [7] as given in (30). Therefore this section will focus on other gauge groups.

A general expression for $\partial_m^2 \hat{Z}_{G_N}^{(k)}|_{m=0}$ is still lacking for gauge groups other than $SU(N)$. So we will be limited to considering particular examples for these cases. Below we will present the expressions for $\widetilde{Z}_{G_N}^{(k)\,gauge}$ and $\widetilde{Z}_{G_N}^{(k)\,matter}$ in (140) for $SO(2N)$, $SO(2N+1)$ and $USp(2N)$, and the prescription of the choice of integration contours, following [19, 20]. From these expressions, explicit results for $\partial_m^2 \hat{Z}_{G_N}^{(k)}|_{m=0}$ are derived and given in section 2.2, which include the one-instanton results for all classical Lie groups as well as some multiple-instanton examples.[19]

Below we list the expressions of $\widetilde{Z}_{G_N}^{(k)\,gauge}$ and $\widetilde{Z}_{G_N}^{(k)\,matter}$ in (140) for the various gauge groups.

- $SO(2N)$

$$\widetilde{Z}_{SO(2N)}^{(k)\,gauge}(m, a_i, \phi_I) = \frac{(-1)^k}{2^k k!} \left( \frac{\epsilon_+}{\epsilon_1 \epsilon_2} \right)^k \frac{\Delta(0)\Delta(\epsilon_+)}{\Delta(\epsilon_1)\Delta(\epsilon_2)} \prod_{I=1}^{k} \frac{4\phi_I(4\phi_I - \epsilon_+^2)}{P(\phi_I + \epsilon_+/2)P(\phi_I - \epsilon_+/2)}, \qquad (141)$$

$$\begin{aligned}
\widetilde{Z}_{SO(2N)}^{(k)\,matter}(m, a_i, \phi_I) = &\left( \frac{(\epsilon_1 + \epsilon_3)(\epsilon_1 + \epsilon_4)}{\epsilon_3 \epsilon_4} \right)^k \frac{\Delta(\epsilon_1 + \epsilon_3)\Delta(\epsilon_1 + \epsilon_4)}{\Delta(\epsilon_3)\Delta(\epsilon_4)} \\
&\times \prod_{I=1}^{k} \frac{P(\phi_I + (\epsilon_3 - \epsilon_4)/2)P(\phi_I - (\epsilon_3 - \epsilon_4)/2)}{(4\phi_I - \epsilon_3^2)(4\phi_I - \epsilon_4^2)},
\end{aligned}$$

where $\epsilon_+ = \epsilon_1 + \epsilon_2$ and $\epsilon_3 = m - \epsilon_+/2, \epsilon_4 = -m - \epsilon_+/2$. The parameters $\epsilon_1$ and $\epsilon_2$ serve as omega deformations to regulate the instanton partition function. The functions $P$ and $\Delta$ are defined as

$$P(x) = \prod_{j=1}^{N} (x^2 - a_j^2), \qquad \Delta(x) = \prod_{I<J}^{k} (x^2 - \phi_{IJ}^2)(x^2 - (\phi_{IJ}^+)^2), \qquad (142)$$

and $\phi_{IJ}^+ = \phi_I + \phi_J$.

The integral is computed by closing the contours in the upper-half complex plan of $\phi_I$, after giving $\epsilon_i$ an imaginary part with the following hierarchy [19]

$$\text{Im}(\epsilon_4) \gg \text{Im}(\epsilon_3) \gg \text{Im}(\epsilon_2) \gg \text{Im}(\epsilon_1). \qquad (143)$$

For the case in which the base manifold is $S^4$ that is relevant for our computation of integrated correlators in $\mathcal{N} = 4$ SYM, we set $\epsilon_1 = \epsilon_2 = 1$, but only after the contour integrals are performed using the prescription described above. This prescription for the choice of contours also applies to the $SO(2N+1)$ and $USp(2N)$ cases that we will discuss next.

---

[19]We would like to thank Francesco Fucito and Francisco Morales for very helpful discussions and for providing their Mathematica code.

- $SO(2N+1)$

$$\widetilde{Z}^{(k)\,gauge}_{SO(2N+1)}(m,a_i,\phi_I) = \frac{(-1)^k}{2^k k!}\left(\frac{\epsilon_+}{\epsilon_1\epsilon_2}\right)^k \frac{\Delta(0)\Delta(\epsilon_+)}{\Delta(\epsilon_1)\Delta(\epsilon_2)} \prod_{I=1}^k \frac{4\phi_I(4\phi_I-\epsilon_+^2)}{P(\phi_I+\epsilon_+/2)P(\phi_I-\epsilon_+/2)},$$

(144)

$$\widetilde{Z}^{(k)\,matter}_{SO(2N+1)}(m,a_i,\phi_I) = \left(\frac{(\epsilon_1+\epsilon_3)(\epsilon_1+\epsilon_4)}{\epsilon_3\epsilon_4}\right)^k \frac{\Delta(\epsilon_1+\epsilon_3)\Delta(\epsilon_1+\epsilon_4)}{\Delta(\epsilon_3)\Delta(\epsilon_4)}$$
$$\times \prod_{I=1}^k \frac{P(\phi_I+(\epsilon_3-\epsilon_4)/2)P(\phi_I-(\epsilon_3-\epsilon_4)/2)}{(4\phi_I-\epsilon_3^2)(4\phi_I-\epsilon_4^2)},$$

with

$$P(x) = x\prod_{j=1}^N(x^2-a_j^2), \qquad \Delta(x) = \prod_{I<J}^k(x^2-\phi_{IJ}^2)(x^2-(\phi_{IJ}^+)^2),$$

(145)

and $\phi_{IJ}^+ = \phi_I + \phi_J$.

- $USp(2N)$

$$\widetilde{Z}^{(k)\,gauge}_{USp(2N)}(m,a_i,\phi_I) = \frac{(-1)^k}{2^k k!}\frac{(\epsilon_+)^{k-\nu}}{(\epsilon_1\epsilon_2)^k}\frac{\Delta(0)\Delta(\epsilon_+)}{\Delta(\epsilon_1)\Delta(\epsilon_2)}\frac{1}{P(\epsilon_+/2)^\nu}\times$$

(146)

$$\times \prod_{I=1}^K \frac{4\phi_I(4\phi_I-\epsilon_+^2)}{P(\phi_I+\epsilon_+/2)P(\phi_I-\epsilon_+/2)},$$

(147)

$$\widetilde{Z}^{(k)\,matter}_{USp(2N)}(m,a_i,\phi_I) = \frac{\left((\epsilon_1+\epsilon_3)(\epsilon_1+\epsilon_4)\right)^{k+\nu}}{\left(\epsilon_3\epsilon_4\right)^k}\frac{\Delta(\epsilon_1+\epsilon_3)\Delta(\epsilon_1+\epsilon_4)}{\Delta(\epsilon_3)\Delta(\epsilon_4)}P((\epsilon_3-\epsilon_4)/2)^\nu\times$$

$$\times \prod_{I=1}^K P(\phi_I+(\epsilon_3-\epsilon_4)/2)P(\phi_I-(\epsilon_3-\epsilon_4)/2)(4\phi_I-(\epsilon_1+\epsilon_3)^2)(4\phi_I-(\epsilon_1+\epsilon_4)^2),$$

where $k = 2K+\nu$ and $\nu = 1$ if $k$ is odd, $\nu = 0$ if $k$ is even. Furthermore, the functions $P,\Delta$ are defined as

$$P(x) = x\prod_{j=1}^N\left(x^2-a_j^2/2\right), \qquad \Delta(x) = \prod_{I<J}^K(x^2-\phi_{IJ}^2)(x^2-(\phi_{IJ}^+)^2)\prod_{I=1}^K(x^2-\phi_I^2)^\nu.$$

(148)

## B.4 One instanton contribution to $\mathcal{C}_{SO(n)}$

Here we consider the large-$y$ expansion, with $y = \pi\tau_2 = 4\pi^2/g_{YM}^2$, of the one-instanton contribution to $\mathcal{C}_{SO(n)}(\tau,\bar{\tau})$ for any $n$, i.e. the perturbation expansion in the one-instanton sector. In the large-$y$ expansion, the one-instanton term can be expressed as

$$\left\langle \partial_m^2\hat{Z}^{(1)}_{SO(n)}(m,a_i)\Big|_{m=0}\right\rangle_{SO(n)} = e^{2\pi i\tau}\left[Y_0(N) + Y_1(N)\frac{1}{y} + Y_2(N)\frac{1}{y^2} + \cdots\right],$$

(149)

where $n = 2N$ or $n = 2N+1$, and the one-instanton contribution to the integrated correlator is given by

$$\mathcal{C}^{(1)}_{SO(n)}(\tau,\bar{\tau}) = \tau_2^2\partial_\tau\partial_{\bar{\tau}}\left\langle \partial_m^2\hat{Z}^{(1)}_{SO(n)}(m,a_i)\Big|_{m=0}\right\rangle_{SO(n)}.$$

(150)

The task is to determine the coefficient functions $Y_i(N)$ in (149).

This is done by expanding $\partial_m^2 \hat{Z}_{SO(n)}^{(1)}(m, a_i)\big|_{m=0}$, as given in (31) for $SO(2N)$ and (34) for $SO(2N + 1)$, in the small-$a_i$ expansion for any $N$ (here we have expanded them to order $a_i^4$). We then take the expectation value according to the matrix model integrals given in (130) and (132) for $SO(2N)$ and $SO(2N + 1)$, respectively. We find the coefficients $Y_i(N)$ obey the following recursion relations:

$$
(2n + 3)(2n + 5)(4n + 9)Y_0(N) - \left(160n^3 + 696n^2 + 728n + 87\right)Y_0(N + 1)
$$
$$
+ 36(n + 1)(n + 2)(4n + 1)Y_0(N + 2) = 0, \tag{151}
$$

$$
(n + 1)(n + 2)(2n + 1)(2n + 3)(4n + 9)Y_1(N)
$$
$$
- 5\left(32n^4 + 56n^3 - 176n^2 - 401n - 183\right)Y_1(N + 1)
$$
$$
+ 36(n - 3)n(n + 1)(n + 2)(4n + 1)Y_1(N + 2) = 0, \tag{152}
$$

$$
(n + 1)(n + 2)(2n - 1)(2n + 1)\left(8n^5 - 6n^4 - 182n^3 - 153n^2 + 333n + 270\right)Y_2(N)
$$
$$
- (n - 1)n\big(320n^7 - 1872n^6 - 5984n^5 + 25526n^4
$$
$$
+ 17178n^3 - 70475n^2 + 1587n + 46962\big)Y_2(N + 1)
$$
$$
+ 36(n - 1)n(n + 1)(n + 2)\big(8n^5 - 86n^4 + 186n^3
$$
$$
+ 155n^2 - 407n + 96\big)Y_2(N + 2) = 0. \tag{153}
$$

These equations apply to both $SO(2N)$ (i.e. using $n = 2N$) and $SO(2N + 1)$ (i.e. using $n = 2N + 1$). Furthermore, the recursion relations can also be solved order by order in $1/n$ expansion, once the initial condition is given. We have used these relations to verify the large-$\tilde{N}_{SO(n)}$ results given in (113).

## C  Laplace-difference equations

In this appendix we will review the evidence for the Laplace-difference equations that hold for any classical gauge group and are summarised in section 3. These equations determine the integrated correlators for any classical gauge group in terms of $\mathcal{C}_{SU(2)}(\tau, \bar{\tau})$, the integrated correlator for the gauge group $SU(2)$.

We begin by reviewing the $SU(N)$ Laplace-difference equations, (40), satisfied by $\mathcal{C}_{SU(N)}(\tau, \bar{\tau})$, which are are described in more detail in [1, 2]. The integrated correlator with gauge group $SU(N)$ has the form (52)

$$
\mathcal{C}_{SU(N)}(\tau, \bar{\tau}) = \frac{1}{2} \sum_{(m,n) \in \mathbb{Z}^2} \int_0^\infty \exp\left(-t\pi \frac{|m + n\tau|^2}{\tau_2}\right) B_{SU(N)}(t)\, dt, \tag{154}
$$

which is the same as (3) with $B_{G_N}^2(t) = 0$ and $B_{SU(N)}(t) \equiv B_{SU(N)}^1(t)$. The function $B_{SU(N)}(t)$ has the form given by (53)

$$
B_{SU(N)}(t) = \frac{\mathcal{Q}_{SU(N)}(t)}{(t + 1)^{2N+1}}, \tag{155}
$$

where $\mathcal{Q}_{SU(N)}(t)$ is a polynomial of degree $2N - 1$ given by (54). In applying the Laplace operator to $\mathcal{C}_{SU(N)}(\tau, \bar{\tau})$ we note the important relation

$$
\Delta_\tau e^{-t\pi Y(\tau, \bar{\tau})} = e^{-t\pi Y(\tau, \bar{\tau})}\left[(\pi t Y(\tau, \bar{\tau}))^2 - 2\pi t Y(\tau, \bar{\tau})\right] = t\, \partial_t^2\left(t\, e^{-t\pi Y(\tau, \bar{\tau})}\right), \tag{156}
$$

where

$$Y(\tau, \bar{\tau}) = \frac{|m + n\tau|^2}{\tau_2}. \tag{157}$$

It therefore follows that applying $\Delta_\tau$ to (154) and after integration by parts, we obtain

$$\Delta_\tau \mathcal{C}_{SU(N)}(\tau, \bar{\tau}) = \frac{1}{2} \sum_{(m,n) \in \mathbb{Z}^2} \int_0^\infty e^{-t\pi \frac{|m+n\tau|^2}{\tau_2}} t \frac{d^2}{dt^2} \Big[ t B_{SU(N)}(t) \Big] dt. \tag{158}$$

To proceed, we note that Jacobi polynomials satisfy the following three-term recursion relation

$$2(n + \alpha - 1)(n + \beta - 1)(2n + \alpha + \beta)P_{n-2}^{(\alpha,\beta)}(z) + 2n(n + \alpha + \beta)(2n + \alpha + \beta - 2)P_n^{(\alpha,\beta)}(z)$$
$$= (2n + \alpha + \beta - 1)\Big[(2n + \alpha + \beta)(2n + \alpha + \beta - 2)z + \alpha^2 - \beta^2\Big]P_{n-1}^{(\alpha,\beta)}(z), \tag{159}$$

as well as

$$(z - 1)\frac{d}{dz}P_n^{(\alpha,\beta)}(z) = n P_n^{(\alpha,\beta)}(z) - (\alpha + n) P_{n-1}^{(\alpha,\beta+1)}(z). \tag{160}$$

From the definition of $B_{SU(N)}(t)$ we find

$$t \frac{d^2}{dt^2}\Big[ t B_{SU(N)}(t) \Big] - 4c_{SU(N)}\Big[ B_{SU(N+1)}(t) - 2B_{SU(N)}(t) + B_{SU(N-1)}(t) \Big]$$
$$- (N + 1)B_{SU(N+1)}(t) - (N - 1)B_{SU(N+1)}(t) = 0. \tag{161}$$

Substituting this relation into (158) gives the Laplace-difference equation (40),

We now turn to the Laplace-difference equations for the integrated correlators of theories with the other general classical gauge groups (41) and (42). Once again the equations are equivalent to differential-difference equations for the rational functions $B_{G_N}^i(t)$ given in subsection 4.2, namely (76) for $B_{SO(2N)}^i(t)$ as well as (79), (83) and (85) for $B_{SO(2N+1)}^i(t)$ and, equivalently, $B_{USp(2N)}^i(t)$.

In the case of $SO(n)$ gauge groups the differential recurrence relation is

$$t \frac{d^2}{dt^2}\Big[ t B_{SO(n)}^i(t) \Big] - 2c_{SO(n)}\Big[ B_{SO(n+2)}^i(t) - 2 B_{SO(n)}^i(t) + B_{SO(n-2)}^i(t) \Big]$$
$$- n B_{SU(n-1)}^i(t) + (n - 1)B_{SU(n)}^i(t) = 0, \tag{162}$$

while for $USp(n)$ (with $n = 2N$) it takes a very similar form,

$$t \frac{d^2}{dt^2}\Big[ t B_{USp(n)}^i(t) \Big] - 2c_{USp(n)}\Big[ B_{USp(n+2)}^i(t) - 2 B_{USp(n)}^i(t) + B_{USp(n-2)}^i(t) \Big]$$
$$+ n B_{SU(n+1)}^{i'}(t) - (n + 1)B_{SU(n)}^{i'}(t) = 0. \tag{163}$$

Note that the rescaling $\tau \to 2\tau$, $\bar{\tau} \to 2\bar{\tau}$ in the second line of (42) implies that, in the above equation (163), $B_{SU(n)}^{i'}(t) = B_{SU(n)}(t)$ when $i = 2$ (as given in (53)), and $B_{SU(n)}^{i'}(t) = 0$ when $i = 1$. Using explicit expressions for $B_{G_N}^i(t)$ given in subsection 4.2, it is straightforward to verify (162) and (163) for any given $N$. Furthermore, the Laplace-difference equations (41) and (42) on the integrated correlators follow from the above equations.

# D   Matching with string theory in $AdS_5 \times S^5/\mathbb{Z}_2$ orientifold

In this appendix we will briefly review the type IIB string theory description that is the holographic dual of the $\mathcal{N} = 4$ SYM theories with classical gauge groups $G_N$.

The holographic equivalence between $\mathcal{N} = 4$ SYM theory and type IIB superstring theory was initially formulated in the context of the $SU(N)$ gauge theory [41–43]. It was argued that in the large-$N$ limit the gauge theory is dual to the string theory in $AdS_5 \times S^5$, which is the near horizon geometry of a stack of $N$ $D3$-branes. According to this correspondence the string coupling is related to the Yang–Mills coupling by $g_s = g_{YM}^2/4\pi$ and the $AdS_5 \times S^5$ length scale, $L$, is related to the RR five-form flux $N$ by $(L/\ell_s)^4 = g_{YM}^2 N$ (where $\ell_s$ is the string length scale). This was soon extended to more general gauge groups and corresponding geometries.

Of particular relevance is the generalisation to theories with general classical gauge groups that still preserve maximal supersymmetry [44, 45]. These are type IIB string theories in an orientifold with background $AdS_5 \times (S^5/\mathbb{Z}_2) \sim AdS_5 \times RP^5$. Such backgrounds emerge from the near horizon geometry of $N$ coincident parallel $D3$-branes that are coincident with a parallel orientifold 3-plane ($O3$-plane). This is the fixed plane of the orientifold projection $\Omega$, which acts on the string world-sheet and the Chan-Paton factors. The fact that the action of $\mathbb{Z}_2$ on $S^5$ is free means that there are no open strings in the type IIB theory in this background and there are also no winding closed strings. The orientifold projection leads to non-orientable string world-sheet contributions in the large-$N$ string perturbation theory obtained from $SO(2N)$, $SO(2N+1)$ and $USp(2N)$ $\mathcal{N} = 4$ SYM. There are four varieties of $O3$-planes that are distinguished by their discrete torsion [45]. This means that they are distinguished by their couplings to $B_{\text{NSNS}}$ and $B_{\text{RR}}$ (the Neveu–Schwarz/ Neveu–Schwarz and Ramond-Ramond two-form potentials), which are flat connections, i.e. $H = dB = 0$, in order to preserve supersymmetry.

The functional integral over a world-sheet $\Sigma$ includes the phase factors $e^{2\pi i \int_\Sigma B_{\text{NSNS}}} = e^{2\pi i \theta_{\text{NSNS}}}$, and $e^{2\pi i \int_\Sigma B_{\text{RR}}} = e^{2\pi i \theta_{\text{RR}}}$ where the torsions $\theta_{\text{NSNS}}$, $\theta_{\text{RR}}$ take the values $0$, $\frac{1}{2}$, and transform as a doublet under $SL(2, \mathbb{Z})$. The various combinations of $O3$-planes that arise are interpreted as follows:

- The orientifold plane with $(\theta_{\text{NSNS}}, \theta_{\text{RR}}) = (0, 0)$ is commonly called the $O3^-$-plane. It is $SL(2, \mathbb{Z})$-invariant and corresponds to the $SO(2N)$ theory. This plane carries $-\frac{1}{4}$ units of five-form RR flux. Together with the $N$ $D3$-branes and their mirror images the total flux of this background is $\tilde{N}_{SO(2N)} = (N - \frac{1}{4})$.

- The other three possible combinations are transformed into each other by $SL(2, \mathbb{Z})$ [45]. The $(0, \frac{1}{2})$ case is the $\tilde{O}3^-$ plane. This is invariant under the self-duality group $\Gamma_0(2)$ and corresponds to the $SO(2N+1)$ theory. The $\tilde{O}3^-$ plane carries $-\frac{1}{4}$ units of flux. However one $D3$-brane is necessarily stuck to it since it coincides with its mirror image. Such a stuck $D3$-brane carries $+\frac{1}{2}$ units of RR flux. Together with the flux of the remaining $N$ $D3$-branes and their mirror images, the total flux in this background is $\tilde{N}_{SO(2N+1)} = (N + \frac{1}{4})$.

- The $(\frac{1}{2}, 0)$ and $(\frac{1}{2}, \frac{1}{2})$ cases are known as $O3^+$ and $\tilde{O}3^+$, respectively. These correspond to the $USp(2N)$ theory in two different duality frames. They are transformed into each other by $\Gamma_0(2)$, which interchanges the monopole states and dyonic states [46]. Since the $O3^+$-plane is a source of $+\frac{1}{4}$ units of RR flux, the total flux in the presence of $N$ $D3$-branes is $\tilde{N}_{USp(2N)} = (N + \frac{1}{4})$.

The relation between the parameters of $\mathcal{N} = 4$ SYM with a general classical gauge group and the length scale in the holographic dual $AdS_5 \times S^5/\mathbb{Z}_2$ is dependent on the RR flux, $\tilde{N}_{G_N}$, of the background. This relation was motivated in [29] by matching the expressions for the trace anomaly in the gauge theory and its holographic supergravity dual, resulting in

$(L/\ell_s)^4 = 2g_{Y M}^2 \tilde{N}_{G_N}$. This generalises the $SU(N)$ gauge theory result and accounts for the values of the strong coupling parameters given in (97).

In the absence of an orientifold projection (i.e. in the large-$N$ $SU(N)$ gauge theory) the world-sheets of string perturbation theory are orientable. The lowest order contribution arises from a spherical world-sheet of order $1/g_s^2$ and the next from a toroidal world-sheet of order $g_s^0$. However, as emphasised in [40, 45] the orientifold projection results in non-orientable string world-sheets that requires the presence of the cross-cap (an $RP^2$ world-sheet), which is of order $1/g_s$ together with non-orientable world-sheets of higher genus. At large $N$ a world-sheet of genus $g$ and $s$ factors of $RP^2$ is of order $N^{2-2g-s}$. Consequently, the large $N$ expansion is an expansion in powers of $1/N$, rather than $1/N^2$. Furthermore, the replacement $N \rightarrow -N$ gives a factor of $(-1)^s$, which clarifies the stringy description of the connection between $SO(2N)$ and $USp(2N)$ theories noted in [47, 48].

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
