# Peer review of "Exact results for duality-covariant integrated correlators in $\mathcal{N}=4$ SYM with general classical gauge groups"

_SciPost Physics, doi:SciPost Phys. 13, 092 (2022)_

## Round 1 · Referee Report · Anonymous (Referee 1) · 2022-4-27

Report

This paper is a continuation of the authors' previous works where an conjectural $SL(2,\mathbb{Z})$-invariant expression for the quantity $\Delta_\tau \partial^2_m \log Z_{S^4}$ of $\mathcal{N}=2^*$ $SU(N)$ theory was found and carefully checked; the novelty in this paper is the extension to other classical gauge groups.

The paper is mostly well-written but the referee wants to request a few improvements.

1) The main recursion relations, (1.12), (3.2) and (3.3) should be given a derivation which should be at least slightly more detailed. The referee understands that the method to be applied was explained in detail for the $SU(N)$ case in the authors' previous papers, but the manuscript in the current form is too terse. Having an extra appendix for the derivation would be nice.

2) The comments around (2.10) and (2.11) are somewhat misguided. It is well-known in the referee's opinion that the large-N expansion of SO and USp gauge theories are given in terms of possibly-unorientable worldsheets. This underlies Maldacena's duality for SO and USp gauge groups, where the holographic dual contains orientifolds. According to the standard review of AdS/CFT by MAGOO, https://arxiv.org/abs/hep-th/9905111 , it goes back to Cicuta, https://inspirehep.net/literature/177713 .

The large N expansion is in terms of $N$ to the power of the Euler number of the worldsheet, and the Euler number is $2-2g$ for an oriented surface of genus $g$, but it can be an odd integer for unoriented worldsheets. Therefore, a better-motivated version of (2.11) would have a summation of the form $\sum_{k} N_G^{2-k}$, where $N_G$ is $N_{SU(N)}=N$, $N_{SO(n)}=n-2$ and $N_{USp(n)}=n+2$, and $k$ is restricted to even integers when $G=SU(N)$. The shift by $\pm2$ does not seem to follow from a group theoretical analysis of the large $N$ Feynman diagrams, but has a natural motivation in AdS/CFT.

What perplexes the referee is that this is essentially reviewed in the Appendix C of this manuscript, and still is not reflected in the comments around (2.10) and (2.11)!

3) The authors commented that the localization computations for the exceptional gauge groups are ill-understood. This is still true to some extent, but it should be noted that there is a way to compute instanton partition functions recursively using something called the blow-up equations, see e.g. https://arxiv.org/abs/1908.11276 . This should allow the authors to check their conjectural lattice-sum formula for $C_G$ by determining $B(t)$ from the perturbative part and comparing its prediction for the instanton contributions against the computations from the blow-up equations. The referee recommends the authors to contact the authors of the paper just mentioned, since setting up the computations using the blow-up equations requires a tedious amount of hard work.

---

## Round 1 · Referee Report · Anonymous (Referee 2) · 2022-6-15

Strengths

  1. The integrated correlators are rich observables, well-deserving of study. The authors give clear and compelling evidence for their intriguingly simple structure as they did for the SU(N) case in earlier work.

  2. The computations are thorough.

  3. The paper is clearly written.

Weaknesses

  1. Some features of the results are obscured.

  2. The paper is a bit long.

Report

Integrated four-point functions in N=4 super-Yang-Mills are, clearly, very intriguing observables, as originally shown by Binder/Chester/Pufu/Wang (ref. [3]) in 2019 for SU(N) SYM and then in subsequent works. There is good motivation to extend previous analyses to the non-simply-laced cases, which these authors do for classical gauge groups.

The results are as expected, for better or worse, but still rewarding. The punchline is that these correlators can be written in essentially similar form, have similar complexity, and obey similar Laplace difference equations which, conjecturally, determine all correlators starting solely from the SU(2) one. The results can be written in a rather uniform way across gauge groups---a very appealing result.

As is often the case, it is less exciting to see the details of the SO(N)/Sp(N) case once the SU(N) case is understood. On the other hand these results will likely be useful for future studies of the SO(N)/Sp(N) SYM theory. For example they should be usable as input in numerical studies of the conformal bootstrap of the SO(N)/Sp(N) theories, following Chester/Dempsey/Pufu's results for the SU(N) theory and Chester's work for SQCD.

The authors stick rather closely to their own blueprint from their earlier works while neglecting some subsequent developments. This could be improved.

A major outstanding question is that it would be nice to say more about the origin of the Laplace difference equation. the Lemma on p.12 is supported by evidence from the perturbative expansions, but for it to carry real weight one would like to know why these relations hold.

The paper deserves publication subject to these remarks.

Requested changes

  1. The work [9] revealed several new things about the SU(N) correlators that ought to have been included in the optimal treatment of the SO(N)/Sp(N) cases. The spectral representation is functionally simpler and makes the meaning of various expressions (for example properties of the kernel, the existence of the lattice representation) physically transparent. This should be more than a sidenote. It is hard to understand why, for example, the (formal) representation as an infinite sum of Eisenstein series with positive integer index receives such emphasis---even in the abstract.

  2. In Sec 4.2 where the integral kernels B are given, the authors explain how to compute them, but not what they are physically. It would have been nice to investigate this. They are presumably best thought of as modified Borel transforms of the perturbative expansion which manifestly preserve the GNO duality?

  3. In the conclusion, the authors mention that it would be nice to find the same lattice representation of the integrated correlator with four mass derivatives. Why would this be expected to exist? Some justification could be provided either way. It seems that actually it may not exist based on previous computations by Chester et al.

---

## Round 2 · Referee Report · Anonymous (Referee 2) · 2022-8-15

Report

This version is better than the last one. It seems fine to publish.

---

## Round 2 · Referee Report · Anonymous (Referee 1) · 2022-8-22

Report

The improvements made by the referee are satisfactory and the paper can be published on SciPost in this referee’s opinion.

---

## Round 2 · Author Response

Thank you for the useful referee reports concerning our paper scipost_202203_00025v1. We have addressed the specific comments made by the referees and are submitting a slightly revised version of the paper to take them into account. The following is a summary of the changes we have made and/or replies to the referees' comments.

---

## Round 2 · List of Changes

{\bf Referee 1} The following describes our responses to the referee's three queries.

1) We have added an appendix, as suggested by the referee, in which we review the Laplace difference equation for $SU(N)$ groups. This appendix is referred to before eq. 3.1. Similarly we discuss the Laplace difference equation for $SO$ and $USp$ groups in the same appendix. In these cases the Laplace difference equations are based on the expressions for the functions $B^1(t)$ and $B^2(t)$ given in section 4.2.

2) We have clarified our comments concerning the perturbative expansion given in (2.11). Whereas in the $SU(N)$ case the 't Hooft expansion parameters are $g_{_{YM}}^2 N$ and $N^2$ at both finite $N$ and large $N$, in the case of the other classical groups the appropriate expansion parameters at finite $N$ are different from the parameters at large $N$. The finite $N$ expansion with small $g_{_{YM}}^2 N$ is very far from the standard AdS/CFT correspondence. The analogue of the '''t Hooft'' parameters in this case are the quantitities $a_{G_N}$ defined in (2.4), while the parameter $N^2$ of $SU(N)$ is replaced by $N_{G_N}$ defined in (2.10). These are different choices from those required at large $N$ and large $\lambda$, which are given in (5.4) and (5.5). These are in accord with AdS/CFT duality, which (as the referee points out) is reviewed in appendix C. They are the unique choices that lead to the remarkable properties of the perturbative expansion (2.5). Although we stressed these properties in the bullett points following (2.9) in response to the referee?s comments we have enhanced the wording in these points since the pattern displayed by the perturbative expansion is one of the main points of the paper. We have also stressed that this expansion is different from the 't Hooft expansion at large N considered in section 4, which can be decomposed into terms corresponding to orientable and non-orientable world-sheets.

3) We have also added a reference to arXiv:1908.11276 in the concluding section, as suggested by the referee.

{\bf Referee 2} The following again describes our responses to the referee's three queries.

1) We think our reference to [9] is appropriate and sufficient, given the content of our paper.. That reference concerned the spectral representation of $SU(N)$ integrated correlators, and is equivalent to our lattice representation presented in [1] and [2] i. For our purposes the lattice representation is somewhat more useful.

This lattice representation was discovered by exploring the perturbative and instanton sectors of the localised correlators defined in [3], which suggested the expression is given by a formal sum of Eisenstein series. In the present paper the key to obtaining the lattice representation for any classical gauge group was again a careful analysis of perturbative contributions (based on [10]) as well as instanton contributions, which are more difficult to determine. This led to the more subtle formal sum of Eisenstein series in (1.8), which satisfy the GNO S-duality constraints and led to the lattice representation (1.3). The expression in terms of Eisenstein series is mentioned in the abstract since it provides a novel mathematical representation of GNO duality that plays a crucial r\^ole in our derivation of the lattice representation.

2) As the referee implies, the functions $B^1$ and $B^2$ were indeed motivated by the perturbation expansion of the integrated correlator. They are constructed in a manner that preserves GNO duality. As described in section 4.3, it is a non-trivial task to extract the instanton data that is consistent with these sums of Eisenstein series. These functions are indeed Borel transforms of the correlator from which the Laplace difference equations and the convergence properties of the correlators in various limits can be determined. The referee asks for an explanation of what $B^1$ and $B^2$ are physically. These expressions emerge from a mathematical analysis of the integrated correlator and do not have an independent physical interpretation.

3) The small comment in the conclusion about the hope of finding a lattice representation of the $\partial_m^4 \log Z|_{m=0}$ correlator is based partly on the large-N expansion explored in Chester et al 2008.02713. There it was shown that the coefficients of integer powers of $1/N$ are sums of 'generalised Eisenstein series' that satisfy inhomogeneous Laplace eigenvalue equations. These do indeed have lattice representations as shown in 0807.0389 and earlier papers. It is not clear to us why the referee suggests that the results in Chester et al suggest the contrary.

---

## Editorial Decision

published